# Geophysical evidence for an enriched molten silicate layer above Mars's core

Henri Samuel[1✉], Mélanie Drilleau[2], Attilio Rivoldini[3], Zongbo Xu[1], Quancheng Huang[4,5], Raphaël F. Garcia[2], Vedran Lekić[5], Jessica C. E. Irving[6], James Badro[1], Philippe H. Lognonné[1], James A. D. Connolly[7], Taichi Kawamura[1], Tamara Gudkova[8] & William B. Banerdt[9]

The detection of deep reflected S waves on Mars inferred a core size of 1,830 ± 40 km (ref. 1), requiring light-element contents that are incompatible with experimental petrological constraints. This estimate assumes a compositionally homogeneous Martian mantle, at odds with recent measurements of anomalously slow propagating P waves diffracted along the core–mantle boundary[2]. An alternative hypothesis is that Mars's mantle is heterogeneous as a consequence of an early magma ocean that solidified to form a basal layer enriched in iron and heat-producing elements. Such enrichment results in the formation of a molten silicate layer above the core, overlain by a partially molten layer[3]. Here we show that this structure is compatible with all geophysical data, notably (1) deep reflected and diffracted mantle seismic phases, (2) weak shear attenuation at seismic frequency and (3) Mars's dissipative nature at Phobos tides. The core size in this scenario is 1,650 ± 20 km, implying a density of 6.5 g cm$^{-3}$, 5–8% larger than previous seismic estimates, and can be explained by fewer, and less abundant, alloying light elements than previously required, in amounts compatible with experimental and cosmochemical constraints. Finally, the layered mantle structure requires external sources to generate the magnetic signatures recorded in Mars's crust.

Mars, like other differentiated terrestrial planets, is composed of an iron alloy core overlain by a silicate mantle and a crust. This metal–silicate dichotomy probably originates from an early global magma ocean stage[4,5] during which heavy iron gravitationally separates from the lighter silicates to form a core[6]. Observations collected by space missions have improved our knowledge on Mars's present-day internal structure. Among these, the NASA InSight mission[7] that deployed the first seismometer on the surface of Mars has been instrumental[8]. Quake recordings[9,10] have determined the layering of the crust[11], its thickness[12], mantle structure[13–15], the core size of Mars and its composition[1,16,17]. The seismic detection of Mars's core confirmed the geodetic measurements[18–21], with an inferred core radius of $R_c = 1,830 ± 40$ km (ref. 1), recently revised to slightly smaller values ($R_c = 1,780 – 1,810$ km)[17], and implies a relatively low core density of approximately 6.0–6.2 g cm$^{-3}$. This core size was determined by the detection of S waves reflected at a solid–liquid interface located at the bottom of the solid mantle, and ascribed to be the core–mantle boundary (CMB). Like all InSight pre- and post-mission structure models of Mars, the mantle was assumed to be compositionally homogeneous[14,17,22–26].

The presence of alloying light elements can explain a low core density, though cosmochemical and experimental constraints do limit this explanation. Sulfur (S) alone could produce the required densities, but with concentrations significantly above the maximum allowable by cosmochemistry[27]. Hence, additional elements have been invoked to

reduce the amount of S, with oxygen concentrations around 5 wt% and carbon and hydrogen concentrations of approximately 1 wt% (ref. 1). However, such concentrations of hydrogen and carbon are significantly higher than allowed by experimental observation. Indeed, experimental constraints[28] limit the hydrogen content in the core to 0.15 wt%. In addition, sulfur decreases the solubility of carbon in Fe-rich alloys from more than 4 wt% (for an S-free core) to below 1 wt% (ref. 29) for a core containing 16 wt% of sulfur. Finally, oxygen concentration in molten Fe–S alloys is controlled by the sulfur content[30], and cannot be larger than 4 wt%, a value well below that proposed in ref. 1.

Since the first seismic determination of Mars's present-day structure[1,13,14,24], additional events have been recorded by InSight. In particular, on Sol (a Martian day, whose duration is 24 h 40 min) 1,000 after InSight's landing, a seismic event triggered by a meteorite impact (hereafter named S1000a) was detected. The impact was precisely located by orbital imaging at 125.9° away from InSight[2], and triggered seismic signals interpreted as the first-ever observation of P waves diffracted along Mars's CMB (hereafter labelled Pdiff)[31]. Among hundreds of models published[1,14,24,32] only a handful of models can marginally fit the measured differential arrival times between Pdiff and PP (that is, a P wave reflected once at the surface of the planet)[2]. A proper fit requires a significant velocity reduction in the deep mantle, which cannot be explained in a homogeneous mantle commonly assumed in Mars's structure models[25,33].

[1]Université Paris Cité, Institut de physique du globe de Paris, CNRS, Paris, France. [2]Institut Supérieur de l'Aéronautique et de l'Espace ISAE-SUPAERO, Toulouse, France. [3]Royal Observatory of Belgium, Brussels, Belgium. [4]Department of Geophysics, Colorado School of Mines, Golden, CO, USA. [5]University of Maryland, College Park, MD, USA. [6]School of Earth Sciences, University of Bristol, Bristol, UK. [7]ETH Zurich, Zurich, Switzerland. [8]Schmidt Institute of Physics of the Earth, Russian Academy of Sciences, Moscow, Russia. [9]Jet Propulsion Laboratory, California Institute of Technology, Pasadena, CA, USA. ✉e-mail: samuel@ipgp.fr

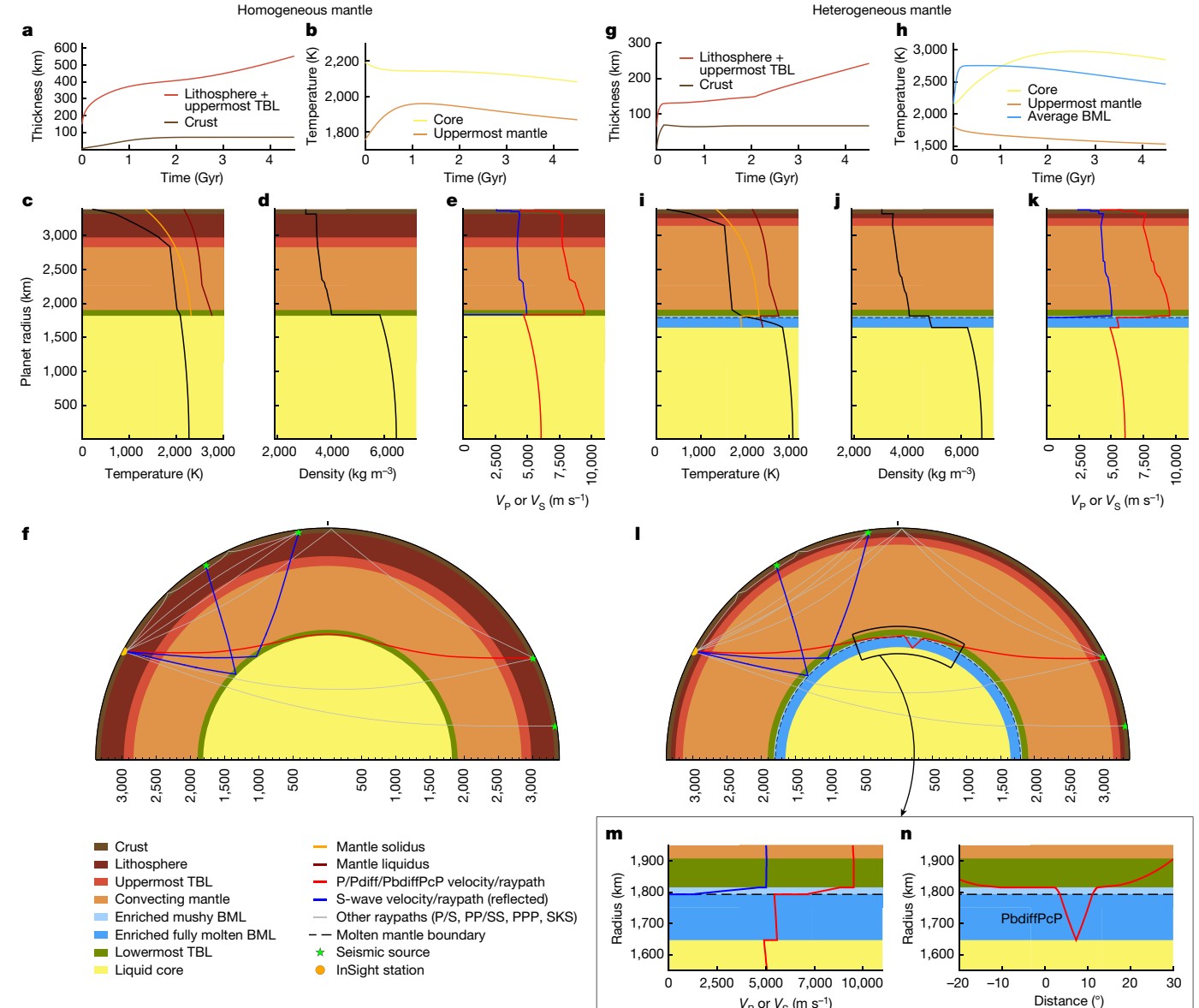

**Fig. 1 | Inversion results.** Thermochemical evolution and present-day structure of Mars. One model among the best 50 is displayed for each inversion set. **a–f**, Without a BML (homogeneous mantle), with $\eta_0 = 6 \times 10^{21}$ Pa s, $E^* = 300$ kJ mol$^{-1}$, $V^* = 3.8$ cm$^3$ mol$^{-1}$. **g–l**, With a BML (heterogeneous mantle), with $\eta_0 = 5 \times 10^{20}$ Pa s, $E^* = 110$ kJ mol$^{-1}$, $V^* = 4.4$ cm$^3$ mol$^{-1}$. **a,g**, Evolution of crustal and lithospheric thicknesses (including the uppermost mantle thermal boundary layer (TBL)). **b,h**, Evolution of uppermost convective mantle ($T_m$) and core ($T_c$) temperatures. **c,i**, Present-day temperature profiles and mantle melting curves from ref. 48 accounting for the influence of iron in the BML[3]. **d,j**, Density profiles. **e,k**, Shear and compressional wave speed profiles. **f,l**, Raypaths for waves reflected at (blue) or diffracted along (red) deep mantle interfaces. Additional raypaths for other phases considered for the inversion are shown in grey. **m,n**, Close-up views of the region delineated in the vicinity of the BML in **l**, showing the P- and

S-wave velocity structure (**m**) and raypath (**n**) of the P-diffracted wave reflected at the CMB (PbdiffPcP). In the homogeneous mantle, S-wave reflection occurs at the CMB, while in the heterogeneous mantle, it occurs above the CMB where velocity decreases abruptly due to the transition from a partially molten to a fully/essentially molten state in the BML (dotted curves). In the heterogeneous mantle, the P-diffracted phase (PbdiffPcP) travels in a molten silicate mantle with slower wave speeds compared with those in a solid mantle, significantly delaying its travel time. The PbdiffPcP phase results from multiple rays diffracted at the top of the fully molten BML before and/or after core reflection, which contribute to this seismic phase. The path displayed corresponds to one of these contributions, which is the reason why it is not symmetric. However, because the seismic model is spherically symmetric, the sum of the contributions will result in a symmetric path (Supplementary Fig. 4).

Additionally, the detection of distant small magnitude events by InSight suggests that seismic attenuation is weak, with effective shear quality factors in excess of 1,000 (ref. 9). This contrasts with the attenuation measured at larger periods such as during Phobos's main tides (5 h 55 min), with a corresponding global quality factor $Q_2 \cong 95 \pm 10$ (refs. 34,35) indicative of a relatively attenuating mantle for tides. This behaviour may be explained by a stratified mantle, akin to a soft, deep mantle layer, as proposed for the Moon[36].

The validity of the common assumption of a homogeneous Martian mantle is brought into question by seismic, geodetic, cosmochemical and experimental observations. In fact, the solidification of Mars's early magma ocean could produce a heterogeneous mantle, leading to a deep-rooted silicate layer strongly enriched in iron and heat-producing elements (HPEs) just above the core[37–41], which is also suggested by isotopic anomalies measured in Martian meteorites[42–44]. Such a basal mantle layer (hereafter BML) can strongly affect Mars's thermochemical

evolution, and may considerably influence the interpretation of available geophysical data[3,33]. Its presence leads to the development of a soft, partially molten mantle layer that could explain Mars's dissipating behaviour for tides[3]. It also implies that the observed S-wave reflections occur near the top of the BML and not at the CMB, suggesting a smaller and denser core than the recently inferred values, which appears compatible with the recently measured nutation of Mars[45]. These lines of evidence suggesting the existence of a previously unrecognized BML motivate the reinterpretation of available data used to constrain the interior structure of Mars. To determine planet structures compatible with observations, we performed a probabilistic inversion of seismic data (Supplementary Information Sections 1 and 3). We considered a non-BML inversion set, with a compositionally homogeneous mantle, as commonly assumed for Mars, and a second set including a BML above the core[3]. Our inversion is parameterized in terms of quantities that influence the thermochemical evolution of the planet composed of a liquid iron core, a silicate mantle (with or without a BML), and an evolving lithosphere and crust[26,46]. Mantle viscosity controls the thermal evolution of the planet and depends on temperature, $T$, and pressure, $P$ (for example, ref. 47):

$$\eta(r) = A_{\text{ref}}\ \eta_0 \exp\left(\frac{E^* + P(r)\ V^*}{R\ T(r)}\right), \qquad (1)$$

where the effective activation energy $E^*$ expresses the temperature sensitivity, the effective activation volume $V^*$ controls the pressure dependence, $R$ is the gas constant, $\eta_0$ is the reference viscosity and $A_{\text{ref}}$ is a prefactor.

Figure 1 underlines the differences between standard (Fig. 1a–f) and BML models (Fig. 1g–l). For a homogeneous mantle, core cooling is monotone, while the uppermost mantle temperature ($T_m$) first increases due to HPEs, before decreasing again. The hot (approximately 1,760 K) initial mantle temperatures favour a rapid crustal growth that depletes the mantle in heat sources, leading to mantle cooling (Fig. 1a,b). This yields a $T_m$ approximately 100 K hotter than initially (Fig. 1b). The present-day temperature profile (Fig. 1c) results in mineralogies and associated density (Fig. 1d) and seismic velocity profiles (Fig. 1e). The latter lead to S-wave reflections occurring at the CMB, as in ref. 1, and a Pdiff phase propagating along the CMB (Fig. 1f).

The BML acts as a buffer that reduces heat transfer between the core and the mantle, but also as a heat source[3], resulting in substantial core heating from approximately 2,160 K to approximately 2,840 K (Fig. 1h). Since the BML segregates most of the HPEs, heat sources are depleted in the overlying mantle, leading to a rapid mantle cooling from approximately 1,820 K to approximately 1,530 K. Yet, the mantle is initially about 60 K hotter than in the homogeneous model (Fig. 1g–l). This favours shallow mantle melting and low mantle viscosities (equation (1)), leading to more efficient cooling, a crustal formation completed more than 500 Myr earlier, and a lithospheric growth about two times slower (Fig. 1g) than the homogeneous case (Fig. 1a). The BML HPE enrichment leads to temperatures exceeding melting curves[48], resulting in a present-day BML fully and partially molten, overlain by a cold ($T_m$ approximately 1,530 K) depleted mantle (Fig. 1i). This present-day temperature profile leads to a distinct density structure and faster velocities in the mantle above the BML but strongly reduced P- and S-wave velocities ($V_P$ and $V_S$) in the hot and molten BML (Fig. 1j–k). The zero $V_S$ in the lower part of the BML yields an S-wave reflection occurring 150 km above the CMB, at the interface where the melt fraction becomes large enough to behave as a liquid material[49,50] (Fig. 1l). In this case, there exists a diffracted P wave in the deep mantle, whose path differs significantly from the one associated with the non-BML case (Fig. 1f). This P wave diffracts along the bottom of the depleted mantle just above the BML, then travels down through the BML before reflecting back at the CMB towards the surface (Fig. 1l–n). We call this core-bouncing diffracted phase PbdiffPcP (whose occurrence is

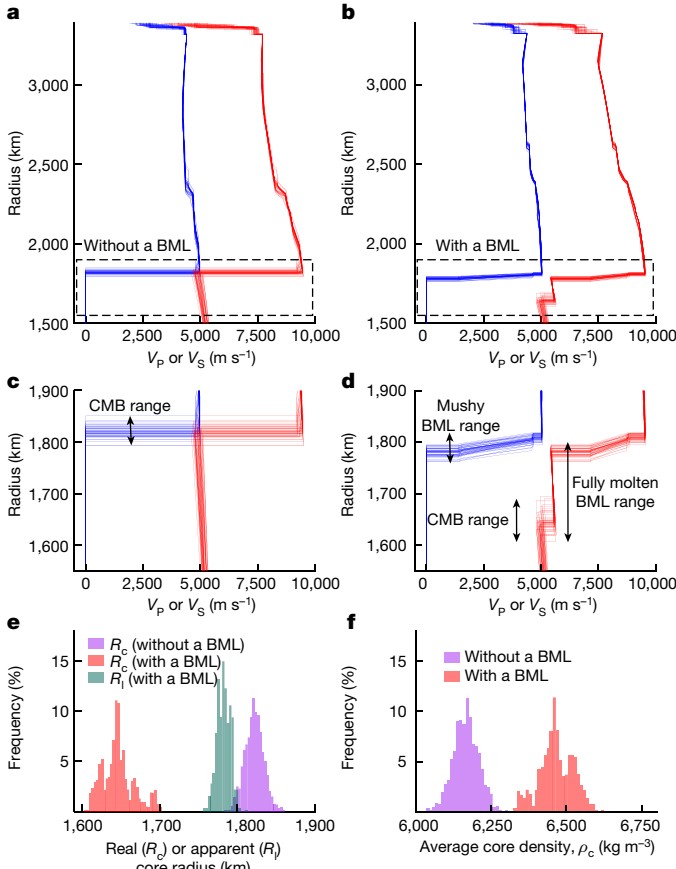

**Fig. 2 | Inversion results of Mars's seismic data with or without a BML.**
**a**, Seismic profiles for P-wave (red) and S-wave (blue) velocities for 50 models chosen among the best 1,000 models without a BML. **b**, Same as **a** but for a mantle that contains a BML. **c,d**, Zoom-in of the areas delineated by the dashed rectangles in **a** and **b**, respectively. The maximum or minimum depth ranges of three distinct seismic regions and interfaces are marked by vertical arrows: the CMB range, the range for the interface between the mushy layer and the fully molten BML, and the fully/essentially molten BML region. **e**, Histograms for real and apparent core radii for the best 1,000 models. The core radius is considerably smaller when a BML is present but the apparent core radii (that is, the radius of liquid iron alloy plus the thickness of the fully molten silicate layer) are similar in both cases. **f**, Histograms of Mars's core density for the best 1,000 models. The smaller core size in the heterogeneous-mantle case leads to a denser core compared with the homogeneous-mantle case. Panels **a**–**d** contain a smaller number of models compared with **e** and **f** to allow for a clear visualization of the seismic structures.

demonstrated in Supplementary Information Section 2). The PbdiffPcP wave travels within the molten silicate layer (Fig. 1m,n), thereby increasing its travel time significantly (Fig. 1k).

Further differences emerge between BML and non-BML sets (Extended Data Table 1). The present-day mantle including a BML is approximately 300 K colder because the mantle is, on average, initially 70 K hotter and about 30 times less viscous than non-BML models. In addition, BML models have $E^*$ and $V^*$ about three times smaller and about 60% larger, respectively, than those associated with non-BML models. The combination of a cold mantle with a small $E^*$ for the BML set leads to uppermost present-day mantle viscosities comparable to those in homogeneous models ($\eta \cong 10^{21}$ Pa s). The differences in $V^*$ imply that the convecting mantle (regions displayed in orange in Fig. 1) above the BML becomes considerably more viscous. Finally, the initially hotter mantle, but now colder in the present-day, results in a thinner lithosphere for the BML set.

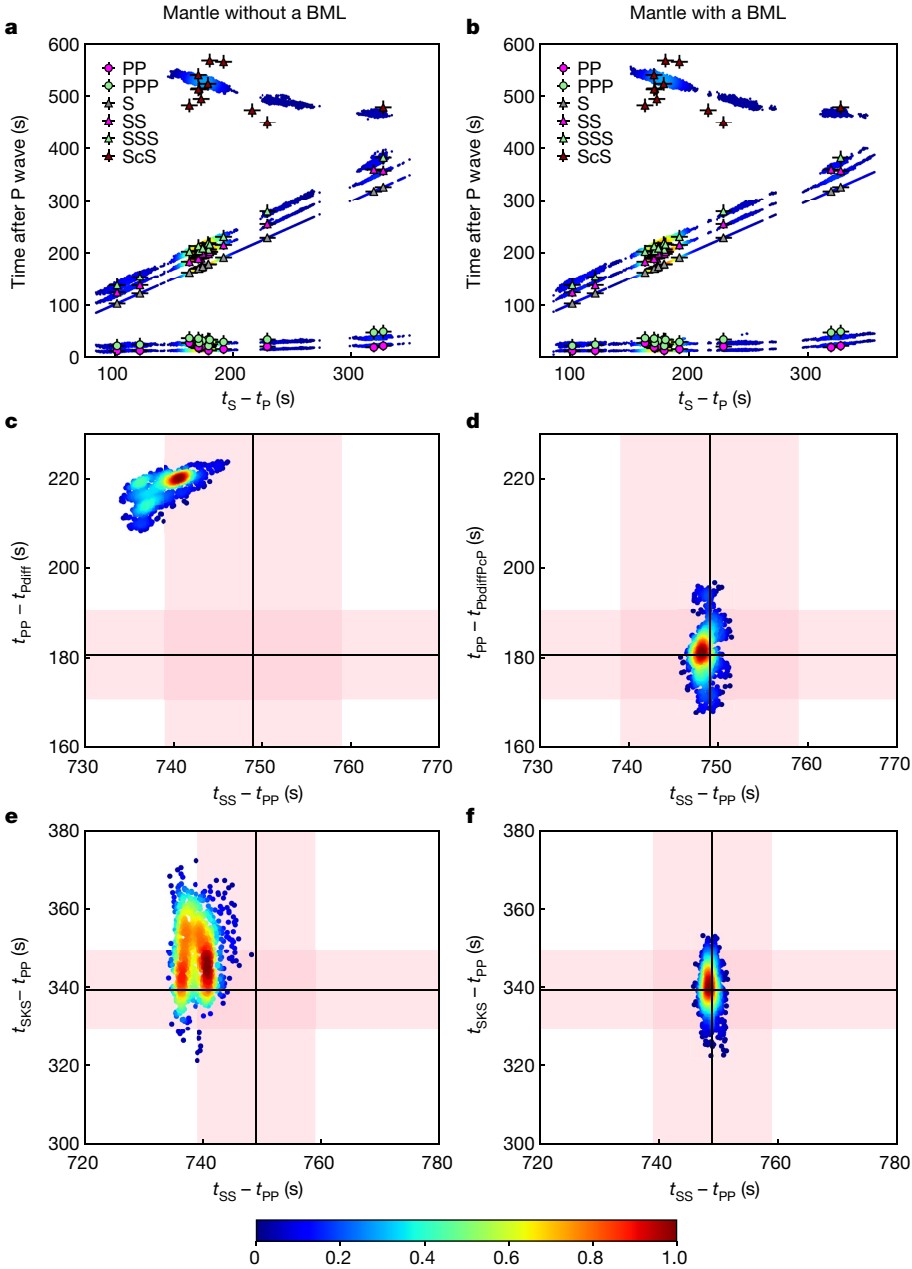

**Fig. 3 | Data fit for the differential travel times, considering a mantle without and with a BML for the best 1,000 models.** The results are displayed in terms of probability density functions (PDFs). Blue and red colours show small and large probabilities, respectively. **a**,**b**, Body wave differential times with respect to P waves as a function of $t_S - t_P$. Markers and error bars represent observed differential arrival times and their uncertainty, respectively. **c**–**f**, $t_{PP} - t_{Pdiff}$ (**c**), $t_{PP} - t_{PbdiffPcP}$ (**d**) and $t_{SKS} - t_{PP}$ (**e**,**f**) as a function of $t_{SS} - t_{PP}$ for event S1000a. The observed differential travel time measurements are displayed with black lines. The pink bands indicate uncertainties on the measurements. Panels **a**,**c**,**e** correspond to the inversion set without a BML, and **b**,**d**,**f** to the inversion set with a BML.

Core quantities also differ between the two inversion sets. For the BML set $R_c$ is approximately 170 km smaller ($R_c = 1,650 \pm 20$ km versus $R_c = 1,820 \pm 10$ km) and therefore denser (average core density $\rho_c = 6,470 \pm 60$ kg m$^{-3}$ versus $\rho_c = 6,130 \pm 30$ kg m$^{-3}$) than the non-BML set to satisfy constraints on the mass of Mars. The CMB temperature is approximately 700 K hotter for the BML set.

While the seismic structures for models with homogeneous mantles simply consist of a liquid core with $V_S = 0$ m s$^{-1}$ overlain by a solid silicate envelope (Fig. 2a,d), the seismic structures for the BML models are more complicated. The increasing temperature with depth (Fig. 1i) causes an increase in melt fraction with depth in the BML. Therefore, the layer is

subdivided into a mushy part (that is, with a melt fraction below 65% (ref. 50), implying strongly reduced but non-zero $V_S$) overlying a fully/ essentially molten part (that is, with $V_S = 0$ m s$^{-1}$) in various proportions (Fig. 2d). The core radius plus the thickness of the fully molten silicate layer (that is, the apparent core radius, $R_l = 1,780 \pm 20$ km) is comparable, yet approximately 40 km smaller than the core size for the non-BML inversion set. However, $R_l$ leans towards recent re-estimation of $R_c = 1,780 - 1,810$ km (median range) using a larger dataset[17].

Constraints from S waves reflected off a deep solid–liquid interface yield a core on average 170 km smaller than the homogeneous case (Fig. 2e). Yet, the BML apparent radii are comparable to $R_c$ in the

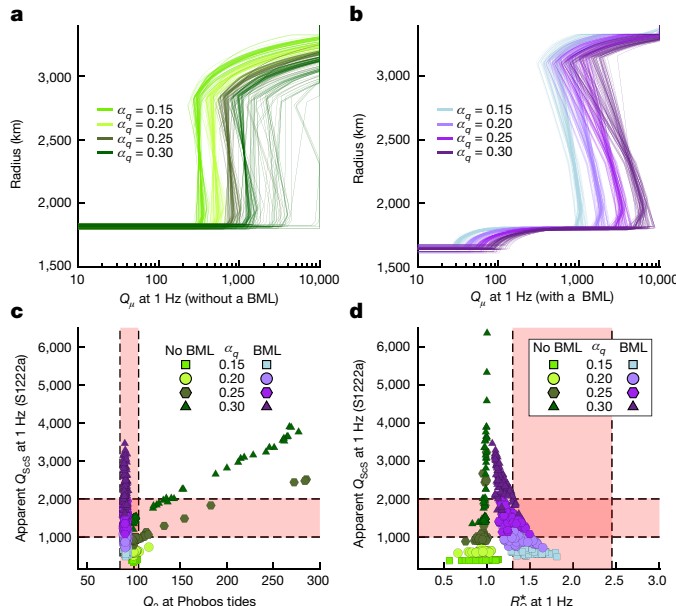

**Fig. 4 | Attenuation at seismic and Phobos's tide periods deduced from the 100 best models. a,b**, Present-day shear attenuation profiles at 1 Hz for the inversion set without a BML (**a**) and with a BML (**b**) for various frequency dependence ($\alpha_q$) of the shear quality factor. **c**, Apparent shear quality factor for the S-reflected wave, $Q_{ScS}$, at 1 Hz and for event S1222a as a function of the global quality factor at Phobos tidal frequency (5 h 55 min). $Q_{ScS}$ was computed by integrating $1/Q_\mu$ for each model along the ScS raypath, determined using the TauP toolkit[60]. Models without BML are displayed in green tones and models with a BML are as shown in purple tones depending on the value of $\alpha_q$. The acceptable ranges for $Q_{ScS}$ and $Q_\mu$ are displayed in pink. **d**, $Q_{ScS}$ at 1 Hz and for event S1222a as a function of $R_Q^*$, the ratio of reflected S to direct S amplitudes $A_{ScS}/A_S$ between events S0185a and S1222a at 1 Hz that samples different solid mantle depths. The colour and symbol coding are identical to that in **c**.

homogeneous cases, due to travel time constraints from deep reflected S waves. As seen above, the smaller core in the BML set has a higher density than homogeneous models (Fig. 2f). In agreement with ref. 45 this BML density structure is compatible with constraints from Mars's core nutation (Supplementary Information Section 6), and models with large $R_1$ display the smallest misfit to observations[51].

While both inversion outputs generally exhibit good fit to seismic data (Fig. 3a,b), the BML models specifically yield a considerably better fit to waves propagating in the lowermost mantle and in the core. For a homogeneous mantle, $t_{PP} - t_{Pdiff}$ falls outside the 1$\sigma$ uncertainty (Fig. 3c), confirming that the P wave diffracted along the CMB travels too fast relative to the PP-phase[2]. By contrast, the molten BML layer significantly delays the PbdiffPcP propagation, leading to $t_{PP} - t_{PbdiffPcP}$ satisfying seismic observations (Fig. 3d). Additionally, the BML seismic structure generates waveforms comparable to InSight's seismic record (Supplementary Information Section 4).

Shear quality factor profiles, $Q_\mu(r)$, can be extracted from our inversion output. Similar to viscosity, $Q_\mu$ depends on temperature and pressure, and is a function of frequency, $\omega$ (refs. 23,52,53): $Q_\mu(r, \omega) \propto [\eta(r)/\omega]^{\alpha_q}$, where the power-law exponent $\alpha_q \cong 0.1$–0.3 expresses the frequency dependence of $Q_\mu$ (refs. 23,35). The BML $Q_\mu$ profiles at 1 Hz (Fig. 4b) have low values in the partially molten mantle. Elsewhere, $Q_\mu$ is relatively large for both inversion outputs, indicating weak seismic attenuation in the solid mantle. In this region, BML models exhibit a steeper increase in $Q_\mu$ (Fig. 4a–b). Deep mantle seismic $Q_\mu$ can also be inferred using the relative attenuation between S and ScS phases to infer $Q_{ScS}$, the apparent $Q_\mu$ along an S wave reflected in the lowermost mantle. The approach relies on the amplitude ratios for ScS and S phases (Methods and Supplementary Information Section 5),

yielding $Q_{ScS} = 1{,}500 \pm 500$. This contrasts with the relatively attenuating Martian mantle at longer tidal periods: $Q_2 = 95 \pm 10$. While BML and non-BML models satisfy the constraints on $Q_2$ (Fig. 4c), several non-BML models with large $\alpha_q$ fall outside the range $Q_2 = 95 \pm 10$ while matching $Q_{ScS} = 1{,}500 \pm 500$ (Fig. 4c). This reflects the difficulty of satisfying geodetic and seismic constraints across a broad range of time scales. BML models are less affected because the partially molten layer accommodates tidal dissipation[3].

An additional observational constraint can be gained by considering the gradient of the apparent $Q_\mu$ in the solid convecting mantle, $R_Q^*$, with $R_Q^* < 1$ indicating a depth decreasing $Q_\mu$, and vice versa. We estimated $R_Q^*$ from the travel times and $Q_{ScS}$ for events sampling different mantle depths (Methods), leading to $R_Q^* = 1.9 \pm 0.6$. Non-BML $R_Q^*$ values are far below this acceptable range, consistent with the weak sensitivity of $Q_\mu$ in these models (Fig. 4a). In contrast, BML models have higher $R_Q^*$ values, with a large fraction matching the acceptable range, consistent with the increase of $Q_\mu$ in the mantle (Fig. 4b). Most models that satisfy both $R_Q^*$ and $Q_{ScS}$ constraints have $\alpha_q = 0.2$–0.25. These results clearly favour BML models, and provide information on the so far poorly constrained frequency dependence of $Q_\mu$ in Mars's solid mantle.

The revised smaller core for BML models is around 5–8% denser than previous seismic estimates, implying a reduced light element content compared to refs. 1,17 (Methods). Indeed, Mars's core can be explained within cosmochemical bounds by a combination of 17 wt% of sulfur and 2.9 wt% of oxygen in addition to iron, or by a more complex mixture of sulfur, oxygen, hydrogen and carbon in various smaller proportions (Methods and Supplementary Information Sections 7 and 8).

Mars's magnetic activity went extinct 3.8–4.0 Gyr ago[54–56]. Magnetic field production via a thermally driven dynamo action requires efficient convective motion in the metallic core, implying core heat loss at the CMB. However, the BML heating and heat buffering effect prevents core cooling (Fig. 1h). While this suggests that a thermally driven dynamo cannot operate on Mars, alternative external sources may power an early dynamo. These include early core super-heating following core–mantle segregation[57], CMB heat flux enhancement due to the overturn that led to the formation of the BML, dynamo episodes caused by late giant impacts[58], or core elliptical instabilities created by satellites orbiting Mars in retrograde fashion[59]. These mechanisms, alone or combined, may have allowed for a Martian dynamo lasting for several hundred million years[3].

The presence of a BML on Mars leading to the coexistence of fully and partially molten layers above the metallic core requires a reanalysis of geophysical data, in particular InSight's seismic record for deep phases arrival windows, which may reveal a zoology of seismic phases interacting with lowermost mantle discontinuities.

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

## Methods

### Inversion results

Extended Data Table 1 summarizes the inversion results for BML and non-BML sets.

The main and supplementary sets of BML inversions (see Supplementary Information Section 3 for further details) show comparable statistics (see columns 4 and 5 in Extended Data Table 1), with the exception of the P-wave velocity jump across the CMB of opposite sign between the two sets. The velocity jump is negative for the main set and essentially positive for the supplementary set, as a result of the smaller $V_P$ in the liquid silicate layer due to the different equation of state (EoS) for the liquid silicates used in this set. Additionally, the models from the supplementary set are in good agreement with the seismic data and similar to the main BML inversion set (as shown in Fig. 3).

### Attenuation at different frequencies from inversion outputs

Our two sets of inversion outputs can be used to determine the lowermost-to-uppermost viscosity ratio in the solid convecting mantle (that is, the mantle regions excluding the lithosphere and the BML when present). As mentioned previously, the differences in $V^*$ imply that the convecting mantle becomes considerably more viscous for BML models because pressure and temperature gradients in this region are similar for the output models of both inversion sets, but $V^*$ is larger for BML models. Indeed, neglecting the temperature effect (due to the modest adiabatic increase of approximately 130–150 K) compared to that of pressure in the convecting mantle, and assuming a pressure increase in $\Delta P$ of around 15 GPa throughout the convecting mantle, equation (1) implies a lowermost-to-uppermost solid mantle viscosity ratio, $R_\eta \approx \exp[\Delta P V^*/(RT_m)]$, leading to $R_\eta \cong 50$ for non-BML models and $R_\eta \cong 5,000$ for BML models.

The distinct rheology between the homogeneous mantles and BML mantles may not be straightforward to explain because of the remaining trade-offs between poorly constrained quantities that can affect the rheological behaviour (that is, water content, grain size, major element composition). For example, an intrinsically more sluggish mantle can be explained by a drier mantle[47,61] but also by larger grain sizes, which are not explicitly modelled in our inversions.

As for viscosity, our inversion output can be used to deduce shear quality factor, $Q_\mu$, across the planet's radius, $r$. $Q_\mu$ depends on temperature and pressure, and is a function of frequency[23,52,53]:

$$Q_\mu(r, \omega) \cong Q_0 \left[ \frac{1}{\omega} \exp\left( \frac{E^* + P(r)V^*}{RT(r)} \right) \right]^{\alpha_q}, \qquad (2)$$

where the power-law exponent $\alpha_q$ modulates the dependence of attenuation on frequency ($\omega$) and $Q_0$ is a constant adjusted to match constraints on Mars's global degree-two shear attenuation, $Q_2$, at the frequency of the main Phobos tide (5 h 55 min)[53]. We adjusted $Q_0$ to obtain $Q_2$ in the range $Q_2 \cong 95 \pm 10$ (refs. 3,22,34,35). $Q_0$ implicitly contains information on several rheological parameters (that is, reference grain size and exponent or relaxation time scale). While more explicit models could be considered for $Q_\mu(r, \omega)$ (for example, Andrade or Burgers[62,63]) they would display dependencies on $P$, $T$ and $\omega$ (ref. 35) as equation (2) does.

Because the value of the power-law exponent remains debated, with estimates ranging between approximately 0.1 and approximately 0.3 (see refs. 23,35 and references therein), four values of $\alpha_q$ were considered.

In the solid mantle below the lithosphere, BML models exhibit steeper increase in $Q_\mu$ values, with ratios of lowermost-to-uppermost (solid) mantle attenuation of $R_Q = 2$–12, while non-BML models are associated with much smaller values: $R_Q = 1$–3 (Fig. 4a,b). This difference results from the larger values of effective activation volume associated with the BML inversion outputs. Using equations (1) and (2) one can relate the ratios of lowermost-to-uppermost solid mantle viscosities ($R_\eta$) and shear quality factors: $R_Q \approx R_\eta^{\alpha_q}$. Hence, using the previously estimated values of 50 (non-BML) and 5,000 (BML) for $R_\eta$ and for the range of $\alpha_q$ considered, this translates to $R_Q = 2$–3 for non-BML and $R_Q = 4$–13 for BML inversion outputs. These estimates are close to those displayed in Fig. 4a,b. These results can be compared to observational constraints on seismic attenuation within Mars's lithosphere and mantle inferred from InSight's seismic waveforms[64].

To gain information on seismic $Q$ in the deep mantle, one can consider the relative attenuation between S and ScS phases for suitable events to infer $Q_{ScS}$, the apparent $Q$ value along an S wave reflected in the lowermost Martian mantle, without being sensitive to the source cut-off. The latter is proposed to be relatively low for quakes originating from the Cerberus Fossae region[65], which may therefore affect any attenuation measurements. This determination relies on the relative amplitudes $A_{ScS}$ and $A_S$ for ScS and S phases, respectively, recorded by the seismometer (see following section for details). We conducted this analysis using the S1222a event that has the best signal-to-noise ratio and an estimated epicentral distance of 37°, for which S and ScS phases were detected, yielding $Q_{ScS} = 1,500 \pm 500$ associated with this event (see section below).

### Q estimation from direct arrivals

Following ref. 66 we write the amplitude spectrum ($A_\varphi$) of a seismic phase $\varphi$ (for example, S or ScS) as:

$$A_\varphi(\omega) = S(\omega)\varnothing_\varphi \exp[-(\omega t_\varphi)/(2Q_\varphi)], \qquad (3)$$

where $\varnothing_\varphi$ is a constant and represents the radiation pattern and geometric spreading of the $\varphi$ phase. Then, using the above expression we compute the amplitude spectrum ratio between ScS and S as:

$$R_{ScS-S} = \frac{A_{ScS}(\omega)}{A_S(\omega)} = \frac{\varnothing_{ScS}}{\varnothing_S} \exp\left[ -\frac{\omega}{2} \left( \frac{t_{ScS}}{Q_{ScS}} - \frac{t_S}{Q_S} \right) \right]. \qquad (4)$$

This equation expresses the amplitude ratio in theory. Note that the ScS is due to an S-wave reflection occurring at the base of the partially molten BML, below which the S-wave velocity is zero. For an incident Sh wave (like in S0185a), this reflection is a total reflection where all the incident Sh-wave energy is reflected as an Sh wave, and the equation above holds as is. However, for an incident Sv wave (like in S1222a, see Supplementary Information Section 1.2), only a fraction of the incident energy is transmitted to the reflected Sv-wave energy, while the complementary incident energy will be transferred to the reflected P wave or to the P-wave energy in the fully molten layer below the reflector. To characterize the fraction of the incident Sv-wave energy being reflected as Sv waves, we compute the amplitude ratio between the reflected and the incident Sv waves (that is, using the analytical expression for the reflection coefficient, equation 5.40 in chapter 5 in ref. 67) at the base of the partially molten BML using typical values of parameters listed in Extended Data Table 2. In addition to the values listed in Extended Data Table 2, the base of the BML is considered as a sharp boundary and $V_S$ below the boundary is 0 m s$^{-1}$. To avoid numerical instabilities in using equation 5.40 of ref. 67 we replaced the zero value by $10^{-5}$ m s$^{-1}$ for $V_S$ below the BML. We tested other values (for example, $10^{-4}$ m s$^{-1}$ and $10^{-6}$ m s$^{-1}$), which resulted in similar reflection coefficient values compared to those obtained with $V_S = 10^{-5}$ m s$^{-1}$. We found that the reflection coefficient is always larger than 98% for the incident angle ranging from 0 degrees (vertical incidence) to 10 degrees. Thus, we can reasonably neglect these coefficients in equation (4) above.

The estimation of $Q_{ScS}$ using the equation above requires the knowledge of the travel time of ScS and S phases, $Q_S$, and $A_{ScS}/A_S$. For S1222a and S0185a, we picked the S based on the Marsquake Service catalogue and picked the ScS using waveform matching and polarization analysis[1]. We conducted our picking independently, and our S0185a ScS pick

agrees with those in ref. 14. For each event, we used a 20-second-long time window for the two phases after the corresponding arrival times and compute the spectrum ratio in the frequency domain ($A_{ScS}/A_S$). For S0185a, we used the transverse component, as the ScS candidate is obvious on the transverse component; we adopted the S1222a radial component data because the ScS candidate is more pronounced on the radial component. We used $Q_S = 4,000$ from the direct-arrival $Q$ estimation (Supplementary Information Section 6). Analysing this information and the equation above, our estimates of $Q_{ScS}$ range between 1,000 and 2,000. Note that this estimation is essentially model-independent: even though the absolute arrival time $t_S$ is model-dependent we checked that changing its value by 100 seconds did not affect the lower and upper bounds of 1,000 and 2,000 for $Q_{ScS}$. All other parameters that enter equation (4) are based on seismic observations.

Most of the BML and non-BML models satisfy the constraint on $Q_2$. This is not surprising given that $Q_0$ is adjusted to match the required range for $Q_2$ (Fig. 4c). However, several non-BML models with relatively large $\alpha_q$ values do not fall within the acceptable range for $Q_2$ even if they agree with the $Q_{ScS}$ range (Fig. 4c). BML models are less affected because the presence of a deep partially molten layer concentrates the dissipation in this region at Phobos tidal frequency[3]. Note that a partially molten and strongly attenuating layer with a typical thickness of 100 km or less would correspond to only about 10 wavelengths or less for a 2-second-period body-wave, which would therefore not be appreciably damped by the partially molten structure. Furthermore, the presence of the soft basal layer favours a smaller core radius to satisfy constraints on the real part of the tidal Love number, $k_2$, as demonstrated for the Moon[68].

Since a large fraction of BML and non-BML models fall within the acceptable $Q_2$ and $Q_{ScS}$ ranges, it remains difficult to discriminate between the two sets of models because $Q_{ScS}$ values remain dominated by large (shallow) values of $Q_\mu$ along the ScS paths.

Nevertheless, an additional observational constraint can be gained by considering the gradient of the apparent shear attenuation in the solid (that is, convecting) mantle. As seen above, such gradient ($R_Q$) is considerably larger for BML models (Fig. 4a,b). If one considers another event with a larger epicentral distance to compute $Q_{ScS}$, the corresponding S and ScS raypaths will sample deeper regions of the mantle (Fig. 1f,l), which may result in different $Q_{ScS}$ values compared to those associated with S1222a, especially for BML output models that exhibit a larger increase of $Q_\mu$ with depth (that is, larger $R_Q$ values). We applied this approach to event S0185a that produced observable ScS and S phases and has a larger epicentral distance than S1222a (55 degrees for S0185a versus 37 degrees for S1222a), requiring an S-raypath diving into deeper regions of the mantle. Using these two events, the amplitude ratio $R_Q^*$ between $(A_{ScS}/A_S)^{S1222a}$ and $(A_{ScS}/A_S)^{S0185a}$ provides a constraint on the gradient of the shear quality factor in Mars's solid mantle. A ratio $R_Q^*$ smaller than unity indicates a decreasing $Q_\mu$ with depth, and vice versa. Due to the integrated nature of $Q_S$ and $Q_{ScS}$ there is no direct correspondence between $R_Q^*$ and $R_Q$. However, a relative correspondence exists: an increase in $R_Q$ yields an increase in $R_Q^*$, with the converse also being true. One can estimate $R_Q^*$ as a function of travel time and apparent $Q$ for ScS and S phases for events S1222a and S0185a. Making the reasonable assumption, as justified above, that $\varnothing_{ScS}/\varnothing_S \cong 1$, and using equation (4), yields the following expression for relative ratios between the two events:

$$R_Q^* \cong \exp[\omega(\Delta t_S^* - \Delta t_{ScS}^*)/2], \qquad (5)$$

where $\Delta t_\varphi^*$ (with $\varphi$ being either the phase S or ScS) is the difference of the ratios of the travel time to the apparent attenuation, $t_\varphi/Q_\varphi$, between events S0185a and S1222a, which are computed along each corresponding raypath. Note that here the computation of the relative ratios is purely based on the models. The observational range for $R_Q^*$ was determined by fitting independently $R_{ScS-S}$ values obtained in the frequency range 0.2–1.0 Hz using linear expressions. To determine the amplitude ratio $R_{ScS-S}$, we used the same S and ScS time windows from the ScS waveform matching of S0185a and S1222a. We taper the time windows using a cosine function and then convert the tapered waveforms to the frequency domain to obtain the amplitude spectra. For each event, we compute the amplitude ratio between the ScS and S using the corresponding spectra. A few anomalously high peaks, with amplitudes approximately 10 times larger than the $R_{ScS-S}$ values measured for the rest of the population (two peaks for S1222a and one for S0185a), were found at certain frequencies. These peaks are most likely due to noise contamination and were therefore removed before performing the linear fitting. Then, we computed the $L_2$ error norm on the fit for both S1222a and S0185a, leading to $R_Q^* = 1.9 \pm 0.6$.

## Constraints on core composition

The higher core density inferred for BML models implies a reduced content in light elements compared to what was recently suggested in refs. 1,17. The most likely light elements in the core of Mars are, in order of abundance, sulfur (S), oxygen (O), carbon (C) and hydrogen (H)[27]. They are soluble in iron during core formation and abundant enough to affect the elastic properties of the core. To deduce the composition of the core, we first defined the subset of possible Fe–O–S–C–H alloy compositions that are consistent with the nickel and cobalt trace-element composition of the Martian mantle[69] by massively sampling core formation models[70] (Supplementary Information Section 7). Then, we searched for those compositions whose EoS best agrees with the isentropic EoS parameters of the Martian core inferred in this study (density ($\rho_{CMB}$), isentropic bulk modulus ($K_S$) and its pressure derivative ($K'$) at CMB conditions, see Supplementary Information Section 8). The EoS for the liquid alloy was built using the most recent experimental data relevant for the composition, pressure and temperature conditions of the Martian core. This is the most suitable approach to account for conditions prevailing in Mars's core that cannot yet be captured by alternative ab initio approaches (Supplementary Information Section 8). We focused this analysis to BML models only as non-BML models are discussed in previous recent works[1,17]. To deduce the composition of the core we only retain models for which the EoS parameters agree with those of the alloy considered here within 0.5%, 1% and 15% respectively (Supplementary Information Section 8). For all models the average core density can be matched by a Fe–O–S–C–H alloy using different proportion of O, S, C and H, but only a small fraction of those models agree with the acoustic velocity of the alloy (see Supplementary Information Section 8). The revised core size we inferred ($R_c = 1,650 \pm 20$ km) lies at the lower end of the InSight pre-mission range[23], is about 170 km smaller than recent seismically derived estimates and yields an average core density $\rho_c = 6.5$ g cm$^{-3}$, which is about 5–8% denser than previous seismic estimates and can be explained by an iron alloy with a smaller amount of light elements. Contrary to previous recent studies[1,17], the core of Mars can be explained by metal–silicate data within cosmochemical plausible bounds (for sulfur) by a mixture of Fe and 17 wt% of S and 2.9 wt% of O, or by a more complex mixture of S, O, H and C in various smaller proportions. Fe–S–O–C assemblages are most likely composed of 14 wt% of S, 3 wt% of O and 1 wt% of C. The addition of H (in amounts of 0.15 wt%, or even 0.5 wt%, that is, beyond the maximum bound allowed by experimental petrology and cosmochemistry) further decreases the fractions of other light elements. However, the effect of adding H to Fe–S–O–C is, on average, less important (that is, causing a 1 wt% decrease or less of the fractions of S, O and C) because the small quantities of H permitted here do not appreciably affect the proportions of other light elements (Supplementary Information Section 8).

Extended Data Table 3 summarizes the statistics of the results (also displayed in Supplementary Fig. 11), with the addition of a case not shown in in Supplementary Fig. 11, which corresponds to a Fe–O–S–H assemblage where the maximum upper bound for H was set to 0.15 wt%

(as determined in Supplementary Information Section 7.3) instead of 0.5 wt% in the fourth column. Fe–O–S–C assemblages are composed of $15 \pm 2$ wt% of S, $3 \pm 1$ wt% of O and $1 \pm 1$ wt% of C. For the more complex assemblage containing $0.3 \pm 0.1$ wt% of H in addition to the three other light elements, their proportions decrease slightly by 1–2%. Interestingly, an Fe–O–S–H that contains less than 0.15 wt% of H results, on average, in similar composition for the other light elements: $14 \pm 2$ wt% of S, $3 \pm 2$ wt% of O and $1 \pm 1$ wt% of C. These fractions are also similar to those of the simpler assemblage Fe–O–S–C. This is because the small amounts of H allowed do not significantly impact the other light element contents.

## Data availability

The seismic catalogue of Marsquake Service is described in ref. 71. Arrival times and underlying data are provided with this paper and in ref. 14. The metadata used to make the figures can be downloaded at https://doi.org/10.5281/zenodo.8148831 (ref. 72). Source data are provided with this paper.

## Code availability

All the computations made in this article are based on codes described in published papers that are cited in the reference list. The Monte Carlo inversion code is described in refs. 3,26. The travel time and ray tracing code TauP is described in ref. 60 and is available at https://www.seis.sc.edu/taup/. The AxiSEM wave propagation code is described in ref. 73 and is available at https://github.com/geodynamics/axisem. The Perple_X Gibbs free energy minimization code is described in ref. 74 and is available at https://www.perplex.ethz.ch.

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

**Acknowledgements** We acknowledge NASA, CNES, partner agencies and institutions (UKSA, SSO, DLR, JPL, IPGP-CNRS, ETHZ, ICL and MPS-MPG), and the operators of JPL, SISMOC, MSDS, IRIS-DMC and PDS for providing SEED SEIS data. H.S. and M.D. were granted access to the GENCI HPC resources of IDRIS under allocations A0110413017 and A0110412958. Numerical computations were partly performed on the S-CAPAD/DANTE platform, IPGP, France, and on GINZA GA-HI supercomputing facilities. Co-authors affiliated to French laboratories thank the French Space Agency CNES and ANR fund (ANR-19-CE31-0008-08). J.B. acknowledges support from the European Research Council (ERC) under the European Union's Horizon 2020 research and innovation programme (grant agreement no. 101019965-ERC advanced grant SEPtiM). Additional support was obtained from IdEx Université Paris Cité ANR-18-IDEX-0001 for H.S., Z.X., P.H.L., J.B. and T.K. V.L. acknowledges support from NASA grant 80NSSC18K1628 and NASA SSERVI Cooperative Agreement 80NSSC19M0216. J.C.E.I. acknowledges UKSA grant ST/W002515/1. This is InSight Contribution Number 224.

**Author contributions** H.S. and M.D. designed the research. H.S. wrote the main text, with inputs from M.D., J.B., A.R., P.H.L., J.A.D.C. and J.C.E.I. H.S., M.D., A.R., Z.X., Q.H., J.B., P.H.L., R.F.G., V.L., J.C.E.I., J.A.D.C., T.K., T.G. and W.B.B. performed research. H.S. and M.D. wrote and contributed to the Methods section on inversion results. Z.X. and H.S. wrote and contributed to the Methods section on attenuation. A.R., J.B. and H.S. wrote and contributed to the Methods section on core composition. R.F.G., Z.X. and V.L. wrote and contributed to Supplementary Information Section 1. V.L. and J.C.E.I. wrote and contributed to Supplementary Information Section 2. M.D. and H.S. wrote and contributed to Supplementary Information Section 3. M.D. and Q.H. wrote and contributed to Supplementary Information Section 4. T.K. wrote and contributed to Supplementary Information Section 5. A.R. wrote and contributed to Supplementary Information Section 6, with inputs from H.S. J.B. wrote and contributed to Supplementary Information Section 7, with inputs from A.R. and H.S. A.R. wrote and contributed to Supplementary Information Section 8, with inputs from J.B. and H.S.

**Competing interests** The authors declare no competing interests.

**Additional information**
**Correspondence and requests for materials** should be addressed to Henri Samuel.

**Extended Data Table 1 | Summary of the inversion results with a homogeneous mantle and with a BML (best 1,000 output models for each inversion set)**

| Parameter | Meaning | Non-BML | BML (main set) | BML (Suppl. set) | Units |
|---|---|---|---|---|---|
| $E^*$ | Mantle effective activation energy | 320±70 | 120±15 | 120±10 | kJ/mol |
| $V^*$ | Mantle effective activation volume | 4.6±2 | 7.3±2 | 8.1±1 | cm$^3$/mol |
| $\eta_0$ | Mantle reference viscosity | $10^{22\pm0.4}$ | $10^{20.5\pm0.3}$ | $10^{20.8\pm0.5}$ | Pa s |
| $k_d$ | BML thermal conductivity | - | 5±1 | 6±2 | W m$^{-1}$ K$^{-1}$ |
| $T_{m_0}$ | Initial uppermost mantle temperature | 1750±30 | 1820±60 | 1810±70 | K |
| $T_{c_0}$ | Initial CMB temperature | 2230±80 | 2240±75 | 2240±100 | K |
| $T_p$ | Potential temperature | 1790±40 | 1510±20 | 1530±20 | K |
| $T_m$ | Uppermost mantle temperature | 1880±50 | 1540±20 | 1610±130 | K |
| $T_d$ | Average BML temperature | - | 2390±90 | 2310±150 | K |
| $T_c$ | CMB temperature | 2090±40 | 2760±150 | 2570±150 | K |
| $D_{cr}$ | Crustal thickness | 71±1 | 67±2 | 67±3 | km |
| $D_{lu}$ | Lithospheric and upper TBL thickness | 600±70 | 260±20 | 250±10 | km |
| $D_d$ | BML thickness | - | 165±20 | 140±20 | km |
| $R_c$ | Core radius | 1820±10 | 1650±20 | 1660±20 | km |
| $P_c$ | Pressure at CMB | 19±0.2 | 22±0.3 | 21.5±0.3 | GPa |
| $V_{P\,\mathrm{CMB}}^{m}$ | $V_P$ in the mantle at CMB | 9.37±0.09 | 5.59±0.04 | 4.86±0.03 | km/s |
| $V_{P\,\mathrm{CMB}}^{c}$ | $V_P$ in the core at CMB | 4.87±0.12 | 4.99±0.10 | 5.06±0.09 | km/s |
| $\Delta V_{P\,\mathrm{CMB}}$ | $V_P^c - V_P^m$ at CMB | -4.50±0.16 | -0.59±0.11 | 0.20±0.080 | km/s |
| $\rho_{\mathrm{CMB}}$ | Core density at CMB | 5870±40 | 6240±60 | 6240±60 | kg/m$^3$ |
| $K_S$ | Core isentropic bulk modulus at CMB | 140±7 | 160±10 | 160±6 | GPa |
| $K_S'$ | Pressure derivative of $K_S$ | 5.1±0.8 | 6.2±0.5 | 5.7±0.7 | - |
| $\rho_c$ | Average core density | 6130±30 | 6470±60 | 6470±60 | kg/m$^3$ |

Quantities listed correspond to constant or to present-day values unless specified otherwise. The subscripts '0' indicate initial values for quantities that vary in time.

**Extended Data Table 2 | Typical P- and S-waves velocity and density values used to estimate the S-wave reflection coefficient for models with a BML**

| Layer | $V_P$ [km/s] | $V_S$ [km/s] | Density [kg/m$^3$] |
|---|---|---|---|
| Above the BML base | 7.1 | 1.4 | 4800 |
| Below the BML base | 5.5 | 0 | 4800 |

Typical values of $V_p$, $V_s$ and density above and below the BML used to calculate the amplitude ratio between the reflected and the incident Sv-waves at the base of the partially molten BML.

**Extended Data Table 3 | Statistics for the core compositions that match seismic data for four assemblages of light elements considered (average value and 1-sigma range)**

| Element [wt.%] | Fe-O-S | Fe-O-S-C | Fe-O-S-H | Fe-O-S-H (H<0.15 wt%) |
|---|---|---|---|---|
| S | 17.2±0.9 | 14.6±1.7 | 13.0±1.8 | 14.1±1.8 |
| O | 2.9±0.5 | 2.7±0.5 | 2.3±0.5 | 2.6±0.5 |
| C | - | 1.1±0.5 | 1.3±0.6 | 1.3±0.5 |
| H | - | - | 0.3±0.1 | 0.1±0.0 |

The last column corresponds to a Fe-O-S-H with a H content below the absolute maximum bound of 0.15 wt%.