## [Peer Review File · Nature]

Manuscript Title: Geophysical evidence for an enriched molten silicate layer above Mars' core

Reviewer Comments & Author Rebuttals

Reviewer Reports on the Initial Version:

Referee #1:

Review of Evidence and implications of an enriched molten silicate layer above Mars' core

While I am surprised previously published papers by the InSight team on the composition of the Martian core did not take cosmochemical constraints into account, I am glad this paper does.

This is a comprehensive multi-disciplinary paper that utilizes all available constraints (experimental, orbiter observations, cosmochemical, planetary evolution/dynamics, seismology, etc.) to infer an ensemble of models for the structure of the Martian interior that is consistent with all observables. The new model (ensemble) includes a molten Basal Mantle Layer (BML).

With a Basal Mantle Layer (BML) (new model) the core is 180 km smaller (in radius), 700 K hotter, 500 kg/m³ denser, and contains a smaller fraction of S, O, C, and H, while the mantle is 300 K cooler and has a thinner lithosphere and thicker crust than in a model without a BML. The latter model is called "homogeneous mantle model" in the paper, and in this review I will call it "old model", as it is consistent with previously published models.

Expectedly, mantle velocities are slightly higher and allow for a shallow low-velocity zone in the new model. Unexpectedly, core velocities are presented as identical for the bigger, cooler core with a higher percentage of Fe in the new model and the hotter, smaller, less iron-rich core of the old model*. Moreover, the new model contains less volatiles than the old model, which also tends to lower velocities.

* Added note: Fig 2 shows that a P-velocity discontinuity is presented inside the liquid core, 50 km below the boundary between the BML and the liquid core, which is not discussed. This boundary is plotted about 50 km higher (~1700 km) than where the text and other figures suggest it should be (~1640 km?).

The new model is based on comprehensive multi-disciplinary constraints, but also fits seismic data (from InSight) better:

1. The new model increases the time between PP and SS arrivals by 10 s by allowing both to travel well below the thinner lithosphere, while in the old model the SS wave traveled within the high-velocity lithosphere, arriving earlier compared to the PP wave.
2. The new model reduces the time between the Pdiff and the PP arrivals by 40 s by allowing the boundary-grazing Pdiff wave to refract into the low-velocity BML before returning to the mantle and surface and arriving later**.
3. The new model reduced the time between PP and SKS by 10 s by allowing the SKS wave to travel through a slight faster mantle overall, while the PP wave's arrival is experienced a smaller net effect from a thinner lithosphere and cooler mantle.***

** Added note: The CMB-grazing bottom part of the Pdiff wave in Fig 1l is weird. Ray paths that impinge on such a boundary typically refract into the bottom layer while bending away from the normal. So, I expected the Pdiff wave to become a PKP wave, rather than surface where it was observed.

*** Added note: There is no discussion of this in the paper, though there ought to be, given how

important this constraint is. It is also the only constraint on core velocities: Can it be used to better estimate core velocities?

The paper is in pretty great, but not perfect shape. I have some comments about how the presentation of the work can be improved before publication, which I recommend.

A. Equation 2 specifically includes dependence on the radius while equation 1 does not. Yet, eqn 1's dependence on radius is a critical ingredient in the discussion on page 6 on increasing viscosity with depth in the BML model.

B. Page 9 refers to [attenuation in] the solid mantle, while earlier pages refer to the convecting mantle. Do these two terms describe the same part of the mantle?

C. Page 10: "A consequence of a BML on Mars is that its early magnetic field recorded in the crust [58–60] would result from one or several external sources, such as impacts [61], or elliptical core instabilities excited by early satellites orbiting the planet in retrograde fashion [62]. This is because the thermal buffering effect of the BML and its heating due to the enrichment in HPEs results in core heating rather than cooling [3], therefore reducing the duration of a thermal dynamo in Mars' core."

This paragraph is underdeveloped and comes across as an afterthought. It assumes that readers will know what a thermal dynamo is, when it can have been active, and why cooling is required for a planetary dynamo.

D. The discussion on page 4-5 and Fig 1 of mantle evolution with and without a BML is essential but hard to follow due to its qualitative (rather than quantitative) nature (relatively hot, relatively cold, rapid, increases, decreases, depletes, reduces, hotter, shallow, low viscosities). It also vaguely, and likely unintentionally implies that the present-day layering reflects initial layering and that, for example, an initially homogeneous mantle does not evolve into a crust-lithosphere-mantle-BML assemblage.

E. page 6: "However, the differences in V^* imply that the mantle becomes considerably more viscous with depth for BML models because the pressure and temperature gradients in the convecting mantle are similar for the output models of both inversion sets. Indeed, neglecting the relatively small temperature effect compared to that of pressure, and assuming a pressure increase of $\Delta P \sim 15$ GPa throughout the convecting mantle, Eq. 1 implies a lowermost to uppermost solid mantle viscosity ratio, $R \sim \exp[\Delta P V^*/(R T)]$, leading to $R \sim 50$ for non-BML models, and $R \sim 5000$ for BML models."

This is all a bit confusing.

a. E^* being smaller is understandable on account on the lower present-day T for SML models. But why is V^* so much larger in the BML models?

b. "mantle become more viscous with depth": is this in the part of the mantle that is ABOVE the BML?

c. If the p and T gradients are similar in BML and homogenous-mantle models, why does η increase with depth only in the BML model?

d. "small temperature effect compared to that of pressure" Why, where, and when is this happening? Is this related to gradients in E^* and V^* with depth? Is this happening in the mantle between the lithosphere and BML? Is this the case at present or initially?

e. "assuming a pressure increase of $\Delta P \sim 15$ GPa throughout the convecting mantle": An increase from where to where? top to bottom? "throughout" might not be the right word to use.

F. I have three concerns with Figure 1:

1) The core for the BML model (right) is 700 K hotter than the core in the homogeneous model (left), yet the P velocities in the core seem to be the same, as if the bulk modulus and density are

not sensitive to temperature in liquid iron under pressure?

2) The crust for the BML model is 60 km thick and the crust for the homogenous model is 80 km thick. Neither thickness agrees well with those inferred by InSight (30-70 km). In the homogeneous model that is remedied by the bottom part of the crust having seismic velocities that are virtually identical (or very close to) mantle velocities, while still having the same (low) density as the upper crust. This requires a lot of explanation I have not seen in this paper or earlier InSight papers on the crust.

3) "TBL" is used in the figure but not defined.

G. "In addition to seismic arrival times, the detection of distant events of small amplitude by InSight's seismic experiment suggests that seismic attenuation is small, with effective shear quality factors in excess of 1000 [9]."

Do you mean

"In addition to seismic arrival times, the detection of distant events of small MAGNITUDE by InSight's seismic experiment suggests that seismic attenuation is small, with effective shear quality factors in excess of 1000 [9]."

H. "These lines of evidence suggesting the existence of a BML motivate the re-interpretation of available data in the frame of a deep layered Martian mantle."

What is meant with "deep layered mantle"? Do you mean

"These lines of evidence suggesting the existence of a previously unrecognized BML motivate the re-interpretation of available data used to constrain the size of Mars' core."

I. Page 5: "Therefore, from a seismological point of view, the BML acts essentially as an extension of the core for deep reflected S-waves"

This presumes that the core itself is liquid, as suggested in previous publications, but this is not been stated in the current manuscript until the bottom of page 9. It probably needs to be said earlier.

J. Fig 3 shows the 1000 best models and Fig 4 shows the 100 best models. This inconsistency has the appearance or arbitrariness, which might imply there to be no quantitative criteria for model selection.

K. Supplement: The color scales in Figures S1 and S2 are not perceptually uniform, making the images hard to interpret, even for someone who is not colorblind. Please use a perceptually uniform scale, like viridis.

L. Because the seismograms from InSight have a low Signal to Noise Ratio, it is not straightforward to quality-control the arrival times used in the paper within a reasonable amount of time. In addition, most of the arrival times have already been published, hence peer-reviewed, elsewhere.

M. Supplement: The authors state to have used equations from Aki & Richards's chapter 5 to calculate SV reflection coefficients on the boundary between lower mantle and BML. Which equation specifically was or were used? My version seems to be missing the equations for solid-liquid boundaries. And did they use one sharp boundary or cumulative coefficients for the transition layer (partially molten) between lower mantle and the liquid BML?

Referee #2:

I have been asked to review a paper concerning the evidence and implications of a molten silicate layer at the core-mantle-boundary (CMB) of Mars. The paper presents an internal structure model of Mars that has recently been sharpened by the occurrence of impact-driven quakes that generated waves grazing the CMB (S1000a and S0976a). The paper presents conclusions

concerning the thickness of the molten layer, ~150-170 km. The major consequence is that by reducing the liquid core radius, its density increases by ~8%, thus reducing the concentration of light elements (S, C, H, O) required to match the density of molten iron. Such a reduction makes the concentration of light elements required more plausible since it is more consistent with cosmochemical models.

A foundational ingredient of this modelling is the ab initio density and bulk modulus of liquid iron and iron alloys at Mars' core pressures. In this pressure range ($\sim 19 < P < \sim 37$ GPa), the equation of state of liquid or solid iron is not well reproduced by any exchange-correlation functional I am aware of. The agreement improves above 100 GPa at room temperature. Without such reliable essential information, a solid solution model for FeS-Fe, FeO-Fe, Fe₃C-Fe, and for fcc-FeH are developed. The parameters of such models are given in the paper, as well as the density and compressional velocities of these alloys vs. pressure. However, it isn't easy to understand how the core composition modelling was done. One sentence was offered without further elaboration: "Oxygen concentration depends on the concentration of sulfur in the core, the amount of FeO in the mantle, and the conditions of core formation Gendre et al. [2022]..." This sentence is packed with unjustified assumptions. Another sentence: "We, therefore, carried out multi-stage core formation models Badro et al. [2015] scanning all the possible values of input parameters (S content, FeO concentration, magma ocean geotherm, CMB pressure) and determined a range of oxygen concentrations as a function of sulfur concentration." This procedure also needs to be better explained. The authors should give a lot more details on how the compositional models were developed. The uncertainty in the alloys' EoSs is passed on to the core composition, which is modelled to fit the new Martian core model.

The bottom line is: the EoS of iron and light element alloys produced in this paper supports the desired conclusion, i.e., the consistency with cosmochemical constraints.

I will not address other aspects of this paper since they fall outside my expertise.

Referee #3:

This paper explores an enriched partially molten silicate layer at the base of Mars' mantle. The paper presents a nice parallel approach for the two contrasting hypotheses: an inversion with and without the BML is pursued. In general, I find the paper interesting, but the seismic results seem a bit uncertain. Here are some comments/questions:

1. How do the ScS times behave (e.g., if differenced from SS times) for the two models?
2. In considering parsimonious solutions, I would've thought a first approach would be to keep the core the same radius, and partially melt some portion of the lowermost mantle. Could that be done to fulfill differential times, yet still allow ScS to get to the CMB? If the answer is yes, then how is that hypothesis more or less compatible with other data? Wouldn't such a structure result in a negative velocity gradient down to the CMB (but not as severe as models in Figure 2), so wave paths will be strongly affected (and thus times too)?
3. Am I correct to read a +/- 5 sec in reported PP and Pdiff times? (though, the time uncertainty of PP looks huge from Fig S1). If yes, that could be a 10 sec error, correct? That is a significant part of the signal in column 1 of Fig 3 c,e.
4. There are no predictions shown for the seismic waves for the two models, and no ScS waves are shown. That makes confidence in the seismic results difficult because of these uncertainties

In summary, this is a good assessment of the two models. I am not convinced they are the only possibilities, and I struggle with the seismic uncertainties. But it is provocative possibility.

Author Rebuttals to Initial Comments:

Referee #1:

Review of Evidence and implications of an enriched molten silicate layer above Mars' core

While I am surprised previously published papers by the InSight team on the composition of the Martian core did not take cosmochemical constraints into account, I am glad this paper does.

This is a comprehensive multi-disciplinary paper that utilizes all available constraints (experimental, orbiter observations, cosmochemical, planetary evolution/dynamics, seismology, etc.) to infer an ensemble of models for the structure of the Martian interior that is consistent with all observables. The new model (ensemble) includes a molten Basal Mantle Layer (BML).

With a Basal Mantle Layer (BML) (new model) the core is 180 km smaller (in radius), 700 K hotter, 500 kg/m³ denser, and contains a smaller fraction of S, O, C, and H, while the mantle is 300 K cooler and has a thinner lithosphere and thicker crust than in a model without a BML. The latter model is called "homogeneous mantle model" in the paper, and in this review I will call it "old model", as it is consistent with previously published models.

Expectedly, mantle velocities are slightly higher and allow for a shallow low-velocity zone in the new model. Unexpectedly, core velocities are presented as identical for the bigger, cooler core with a higher percentage of Fe in the new model and the hotter, smaller, less iron-rich core of the old model. Moreover, the new model contains less volatiles than the old model, which also tends to lower velocities.*

** Added note: Fig 2 shows that a P-velocity discontinuity is presented inside the liquid core, 50 km below the boundary between the BML and the liquid core, which is not discussed. This boundary is plotted about 50 km higher (~1700 km) than where the text and other figures suggest it should be (~1640 km?).*

The confusion comes from the way the figure was colored in the background, which was misleading. We clarified this figure by removing the color backgrounds and by adding text to better explain the observed seismic structure.

The new model is based on comprehensive multi-disciplinary constraints, but also fits seismic data (from InSight) better:

- 1. The new model increases the time between PP and SS arrivals by 10 s by allowing both to travel well below the thinner lithosphere, while in the old model the SS wave traveled within the high-velocity lithosphere, arriving earlier compared to the PP wave.*
- 2. The new model reduces the time between the Pdiff and the PP arrivals by 40 s by allowing the boundary-grazing Pdiff wave to refract into the low-velocity BML before returning to the mantle and surface and arriving later**.*
- 3. The new model reduced the time between PP and SKS by 10 s by allowing the SKS wave to travel through a slight faster mantle overall, while the PP wave's arrival is experiences a smaller net effect from a thinner lithosphere and cooler mantle.****

*** Added note: The CMB-grazing bottom part of the Pdiff wave in Fig 11 is weird. Ray paths that impinge on such a boundary typically refract into the bottom layer while bending away from the normal. So, I expected the Pdiff wave to become a PKP wave, rather than surface where it was observed.*

Following the reviewer's comment, we carefully re-analysed the path for the Pdiff phase propagating inside the BML, as displayed in Fig. 11 using several complementary forward approaches: full waveform spectral element modelling (using the AxiSEM software), and ray theory (using a fast marching Eikonal solver). We came to the conclusion that the Pdiff path displayed in the original

version of Fig.11 was indeed incorrect. Instead, we found that while P-diffracted paths do exist in the BML structure, this phase diffracts at the top of the BML where the compressional wave speed are maximum, then travels through the partially molten and fully molten BML before being reflected at the CMB back towards the surface instead of being diffracted at the bottom of the BML (see figure below extracted from the revised Fig.1).

Figure: Close-up view of the region in the vicinity of the BML showing the P- and S-wave velocity structure (m) and raypath (n) of the P-diffracted wave reflected at the core-mantle boundary (PbdiffPcP) travels in a molten silicate mantle with considerably slower wave speeds compared to those of a solid mantle, which significantly delays its travel time compared to the homogeneous mantle case.

We have therefore named this phase PbdiffPcP ('b' refers to the BML interface along which this wave first diffracts). We then reassessed the travel time associated with this PbdiffPcP phase, again using spectral element modelling and Eikonal solver, and found consistent arrival times $t_{PbdiffPcP}$. To complete this analysis, we computed $t_{PbdiffPcP}$ using a recent version of the *Taup* software (2.7.0) that properly accounts for this phase, and using the analytical formula (9.31) from Aki & Richards' book and found the same travel times value. Finally, we numerically integrated the PbdiffPcP travel time along the (correct) PbdiffPcP path using the velocity structure from the BML model and found again the same value. Fortunately, it turns out that even though the path for the P-diffracted phase in our BML inversions was wrong, the travel time computed was correct and corresponded exactly to the value of $t_{PbdiffPcP}$ obtained with the other numerical and analytical approaches described above (we systematically checked this for our best 1000 BML models).

The incorrect path was determined with an older version of *Taup* (2.4.5), which however computed the correct travel time for $t_{PbdiffPcP}$ exactly. In fact, by numerically integrating the travel time along the wrong Pdiff path we would find travel times that are systematically about 30 seconds longer than the correct value outputted by *Taup* (obtained with either versions 2.4.5 or 2.7.0). This means that our inversion results, hence the conclusions of our study, are unaffected by this correction.

Consequently, we corrected the ray path in the revised Fig.11 and added the two panels (m, n) to Fig. 1 that display close-up views of the seismic structure and raypath in the vicinity of the BML to better illustrate this finding. Additionally, we devoted a new Section S3 of the supplement to discuss the identification and the existence of the PbdiffPcP phase in the BML context, which completes the full waveform modelling also added to the revised supplement (Section S4).

*** Added note: There is no discussion of this in the paper, though there ought to be, given how important this constraint is. It is also the only constraint on core velocities: Can it be used to better estimate core velocities?

We checked the core velocities between the BML and non-BML sets and found small differences both in terms of values and range. At the top of the core $V_p = 4.87 \pm 0.12$ km/s for non-BML models, while

$V_p = 5.0 \pm 0.10$ km/s. Therefore, the time constraint provided by these phases is similar in both cases. The values mentioned above have been added to the revised manuscript (Table S2).

The paper is in pretty great, but not perfect shape. I have some comments about how the presentation of the work can be improved before publication, which I recommend.

A. Equation 2 specifically includes dependence on the radius while equation 1 does not. Yet, eqn 1's dependence on radius is a critical ingredient in the discussion on page 6 on increasing viscosity with depth in the BML model.

We have now explicitly added the dependence of P and T to the radius r in Equation (1).

B. Page 9 refers to [attenuation in] the solid mantle, while earlier pages refer to the convecting mantle. Do these two terms describe the same part of the mantle?

Yes, they do. We have clarified the terms solid mantle and convecting mantle in the revised version of the manuscript. See also our reply to comment (e) below.

C. Page 10: "A consequence of a BML on Mars is that its early magnetic field recorded in the crust [58–60] would result from one or several external sources, such as impacts [61], or elliptical core instabilities excited by early satellites orbiting the planet in retrograde fashion [62]. This is because the thermal buffering effect of the BML and its heating due to the enrichment in HPEs results in core heating rather than cooling [3], therefore reducing the duration of a thermal dynamo in Mars' core." This paragraph is underdeveloped and comes across as an afterthought. It assumes that readers will know what a thermal dynamo is, when it can have been active, and why cooling is required for a planetary dynamo.

Following the reviewer's advice, we have developed this paragraph in the revised manuscript.

D. The discussion on page 4-5 and Fig 1 of mantle evolution with and without a BML is essential but hard to follow due to its qualitative (rather than quantitative) nature (relatively hot, relatively cold, rapid, increases, decreases, depletes, reduces, hotter, shallow, low viscosities). It also vaguely, and likely unintentionally implies that the present-day layering reflects initial layering and that, for example, an initially homogeneous mantle does not evolve into a crust-lithosphere-mantle-BML assemblage.

We have rewritten this discussion in a more quantitative manner and clarified that the layered planet structure refers to the deep mantle when the BML is present. In non-BML cases, the silicate portion of the planet develops a layered structure but only in the shallow parts with the crust and lithospheric growth. Of course, this also occurs in BML cases but the BML itself leads to additional stratification (*i.e.*, solid, partially molten and liquid units that are enriched in iron and HPEs) in the lowermost mantle. However, the reviewer has the right impression that a non BML mantle at solid-state does not evolve into a BML mantle. It is the homogeneous global silicate magma ocean that can, upon solidification, result in a heterogeneous mantle. However, our approach does not account for magma ocean cooling and for deep mantle differentiation that occurs during the earliest stages of planet formation.

E. page 6: "However, the differences in V^ imply that the mantle becomes considerably more viscous with depth for BML models because the pressure and temperature gradients in the convecting mantle are similar for the output models of both inversion sets. Indeed, neglecting the relatively small temperature effect compared to that of pressure, and assuming a pressure increase of $\Delta P \sim 15$ GPa throughout the convecting mantle, Eq. 1 implies a lowermost to uppermost solid mantle viscosity ratio, $R \sim \exp[\Delta P V^*/(R T)]$, leading to $R \sim 50$ for non-BML models, and $R \sim 5000$ for BML models." This is all a bit confusing.*

a. E^* being smaller is understandable on account on the lower present-day T for SML models. But why is V^* so much larger in the BML models?

Decreasing V^* leads to thinner lithospheres at the present-day, and *vice versa*. Thinner lithosphere would result in slower seismic velocities because in this case, the temperature increases faster with depth in the lithospheric region. The smaller seismic velocities in the lithospheric region would further delay the PP in these BML models. The resulting larger t_{PP} would lead to worse data fit (see Fig. 3). This is the reason why V^* is larger for BML models. We have mentioned this in the revised manuscript.

b. "mantle become more viscous with depth": is this in the part of the mantle that is ABOVE the BML?

Yes, it is. When entering the BML, the viscosity decreases with depth (instead of increasing with depth in the solid mantle) because of the strong temperature gradient in this region. We have clarified that in the revised manuscript.

c. If the p and T gradients are similar in BML and homogenous-mantle models, why does η increase with depth only in the BML model?

The viscosity η increases with depth for both BML and homogeneous mantle models, however, since the activation volume is larger for BML models the viscosity increase with depth (above the BML, see point (b) above) is therefore larger than in the homogeneous mantle models. We clarified this point in the main text.

d. "small temperature effect compared to that of pressure" Why, where, and when is this happening? Is this related to gradients in E^* and V^* with depth? Is this happening in the mantle between the lithosphere and BML? Is this the case at present or initially?

We refer to these changes in viscosity in the convecting mantle at the present day, whose definition is given below (see point (e)). In this region the temperature changes according to the adiabatic gradient (corresponding to a temperature increase of about 130 for the BML models and ~ 160 K for the homogeneous models). Therefore, strictly speaking, the viscosity does change in the convecting mantle. However, the decrease in viscosity with increasing depth in the convecting mantle is much less important than the viscosity increase due to pressure for both models. This is because the change in temperature is smaller than the change in pressure across this entire region.

Indeed, for both BML and non-BML models the viscosity decreases by a factor of $\sim 2-5$ across the convecting mantle (due to the adiabatic temperature increase), while it increases with increasing pressure by a factor $\sim 50-2000$. Therefore, in any case, the viscosity is far more affected by the change in pressure than by the temperature increase in the convecting mantle.

If we had taken the effect of temperature in this reasoning the outcome would have been the same, *i.e.*, $R\eta_{\text{BML}}/R\eta_{\text{homogeneous}} \sim 1000$. We have clarified this in the revised manuscript.

e. "assuming a pressure increase of $\Delta P \sim 15$ GPa throughout the convecting mantle": An increase from where to where? top to bottom? "throughout" might not be the right word to use.

This pressure increase corresponds to the solid mantle that lies below the lithosphere and all the way down to the CMB for homogeneous models and all the way to the top of the enriched layer for BML models, this is what we refer as to convecting mantle because above it, the lithosphere is too stiff to convect and below it there is the core (for homogeneous mantle models) or enriched layer (for BML models), which does not convect either because it is the stably stratified (see Fig. 1f and Fig. 1j for example where the convecting mantle is shown in orange for both homogeneous and BML models). We have further clarified what is meant by convecting mantle in the text by more explicitly referring to the corresponding areas in Fig. 1.

F. I have three concerns with Figure 1:

1) *The core for the BML model (right) is 700 K hotter than the core in the homogeneous model (left), yet the P velocities in the core seem to be the same, as if the bulk modulus and density are not sensitive to temperature in liquid iron under pressure?*

In this figure the relative V_p difference in the core between the homogeneous model (Fig. 1k) and the BML case (Fig. 1e) ranges between 0 and ~3%, which could indeed seem rather small, as the reviewer noted, given the fact that the core temperature is considerably hotter in the BML case. Our models account for temperature dependence of both bulk modulus, K_s , and density, ρ_{cmb} , however these parameters are also inverted for. For the BML case K_s and ρ_{cmb} are larger than in the homogeneous mantle (see Table S2). This increase in both K_s and ρ_{cmb} results in similar V_p values. This convergence towards similar V_p values in the core for both homogeneous and BML models is due to the SKS data that we consider in our inversion, which constrains the value of V_p in the core (hence also K_s and ρ_{cmb}). We have mentioned these points in the revised manuscript.

2) *The crust for the BML model is 60 km thick and the crust for the homogenous model is 80 km thick. Neither thickness agrees well with those inferred by InSight (30-70 km). In the homogeneous model that is remedied by the bottom part of the crust having seismic velocities that are virtually identical (or very close to) mantle velocities, while still having the same (low) density as the upper crust. This requires a lot of explanation I have not seen in this paper or earlier InSight papers on the crust.*

Both the BML and homogeneous models must agree with the average crustal thickness inferred from InSight (ranging between 24 km and 72 km, as the reviewer correctly noted), because this is a requirement from the inversion algorithm, as mentioned in Section 3 of the supplementary information. For the BML model, the crustal thickness at the present day is indeed close to 60 km (precisely 64 km), and therefore lies within the acceptable average crustal mentioned above. The crustal thickness for the homogeneous model is thicker (71 km) but also remains within the 24-72 km range.

3) *"TBL" is used in the figure but not defined.*

We have added the definition of the acronym TBL (Thermal Boundary Layer) both in the figure legend and in the figure caption of Fig. 1 in the revised manuscript.

G. *"In addition to seismic arrival times, the detection of distant events of small amplitude by InSight's seismic experiment suggests that seismic attenuation is small, with effective shear quality factors in excess of 1000 [9]."*

Do you mean "In addition to seismic arrival times, the detection of distant events of small MAGNITUDE by InSight's seismic experiment suggests that seismic attenuation is small, with effective shear quality factors in excess of 1000 [9]."

Yes. We therefore replaced "amplitude" with "magnitude".

H. *"These lines of evidence suggesting the existence of a BML motivate the re-interpretation of available data in the frame of a deep layered Martian mantle."*

What is meant with "deep layered mantle"? Do you mean "These lines of evidence suggesting the existence of a previously unrecognized BML motivate the re-interpretation of available data used to constrain the size of Mars' core."

We modified this sentence following the reviewer's suggestion.

I. Page 5: *"Therefore, from a seismological point of view, the BML acts essentially as an extension of the core for deep reflected S-waves"*

This presumes that the core itself is liquid, as suggested in previous publications, but this is not been stated in the current manuscript until the bottom of page 9. It probably needs to be said earlier.

The assumption of a liquid core is now more clearly mentioned earlier in the main text.

J. Fig 3 shows the 1000 best models and Fig 4 shows the 100 best models. This inconsistency has the appearance of arbitrariness, which might imply there to be no quantitative criteria for model selection.

There was an error in the figure caption. Panels a-d in Fig. 3 display in fact 50 models randomly sampled among the best 1000 models, and not the best 100 models. We corrected the figure caption in the revised manuscript. The reason why we show fewer models in Figs. 3a-d is only for the sake of clear visualization purposes. Superimposing 1000 models in these figure panels would make it difficult to distinguish the seismic structure. We mention this justification in the figure caption of the revised manuscript.

K. Supplement: The color scales in Figures S1 and S2 are not perceptually uniform, making the images hard to interpret, even for someone who is not colorblind. Please use a perceptually uniform scale, like viridis.

We changed the color scale for the two corresponding figures, following the reviewer's suggestion.

L. Because the seismograms from InSight have a low Signal to Noise Ratio, it is not straightforward to quality-control the arrival times used in the paper within a reasonable amount of time. In addition, most of the arrival times have already been published, hence peer-reviewed, elsewhere.

Indeed, most of the arrival times have already been published, which is why we do not present/discuss them further. However, the PP and P-diffracted phases arrival times are discussed in the supplement (Section S1.1), and we further reinforced this discussion in the revised manuscript with the addition of a new figure in Section S1.1.

M. Supplement: The authors state to have used equations from Aki & Richards's chapter 5 to calculate SV reflection coefficients on the boundary between lower mantle and BML. Which equation specifically was or were used? My version seems to be missing the equations for solid-liquid boundaries. And did they use one sharp boundary or cumulative coefficients for the transition layer (partially molten) between lower mantle and the liquid BML?

We computed the S-to-S reflection coefficient by applying Equation (5.40) of Aki & Richards [2002] using values listed in Table S3. In addition to the table, the BML base is considered as a sharp boundary and V_s below the boundary is zero. To avoid numerical instabilities in using Equation (5.40) of Aki & Richards' book we replaced the zero value by 10^{-5} m/s for V_s below the BML. We tested other values (e.g., 10^{-4} m/s and 10^{-6} m/s) that resulted in similar reflection coefficient values than those obtained with $V_s=10^{-5}$ m/s.

Referee #2:

I have been asked to review a paper concerning the evidence and implications of a molten silicate layer at the core-mantle-boundary (CMB) of Mars. The paper presents an internal structure model of Mars that has recently been sharpened by the occurrence of impact-driven quakes that generated waves grazing the CMB (S1000a and S0976a). The paper presents conclusions concerning the thickness of the molten layer, ~150-170 km. The major consequence is that by reducing the liquid core radius, its density increases by ~8%, thus reducing the concentration of light elements (S, C, H, O) required to match the density of molten iron. Such a reduction makes the concentration of light elements required more plausible since it is more consistent with cosmochemical models.

A foundational ingredient of this modelling is the ab initio density and bulk modulus of liquid iron and iron alloys at Mars' core pressures. In this pressure range (~19 < P < ~37 GPa), the equation of state of liquid or solid iron is not well reproduced by any exchange-correlation functional I am aware of. The agreement improves above 100 GPa at room temperature.

We are very confused by this comment, mostly because we fully agree with it. And because we did not use ANY *ab initio* simulations in our paper. Our EoS are based solely on experimental data, which is available in those pressure ranges. Therefore, the foundational ingredient of our modelling is actually experimental (x-ray diffraction at high *P* and *T*) work! This is all detailed in Section 5 of the supplementary information.

Indeed, *ab initio* simulations do a poor job in this pressure range, and this is another reason why we refrained from using them. They do a poor job because, as properly stated by the reviewer, while magnetism in iron decreases with *P* and vanishes at Earth's core pressures, it still plays a role at the pressure of Mars' core and needs to be accounted for. This requires two things: (1) the use of magnetic simulations to correctly estimate the enthalpy of the system, and (2) calculate the magnetic entropy "manually" (because DFT packages like VASP *etc.* will not do it).

A recently published (after our manuscript was submitted) *ab initio* simulation by Huang et al. [2023] (doi 10.1029/2022GL102271) uses magnetic simulations but disregards magnetic entropy. Unsurprisingly, it does not match the existing experimental data previously reported. Instead, the authors recognize that there is a 20% offset in the simulations of pure iron at a single *P* and *T*, and then use an empirical correction to fall back on the experiment; so much for a first principles calculation! Then, they use that same correction for all *P* and *T*, even though entropic effects are notoriously temperature-dependent. And if this isn't bad enough, they apply that same correction of Fe to all Fe-light element alloys such as Fe-S, Fe-O, Fe-C, *etc.* disregarding any compositional effects on magnetic entropy.

Anyway, this is bulletproof evidence that poorly performed *ab initio* simulations should not be used to estimate Martian core properties. Simulations that fully treat magnetism have been performed in the past (*e.g.*, Edgington et al. [2019], doi 10.1016/j.epsl.2019.115838, whose results, despite their sophistication, still show a significant offset from the experimental data) and applied to understand Mercury's core for instance, but the recent paper by Huang et al. [2023] does not go on to such sophistication, which probably explains why it falls short of reproducing experimental data.

To address the reviewer's comment in a single sentence: no, we did not use DFT simulations, which are currently not reliable in this context, and we only used experimental data to establish core thermochemical properties. We clarified this in the revised manuscript.

Without such reliable essential information, a solid solution model for FeS-Fe, FeO-Fe, Fe₃C-Fe, and for fcc-FeH are developed.

We do not use a solid solution model to describe the EoS of the liquid core alloy. All end-member phases are based on EoS deduced from elastic data of liquid alloys. In the previous version of the manuscript, only the FeH end-member was modelled by solid fcc FeH. In the new version we have replaced the fcc FeH EoS by that of liquid FeH recently published by Tagawa et al. [2022]. All details are provided in the Supplement Section 7, as mentioned above.

The parameters of such models are given in the paper, as well as the density and compressional velocities of these alloys vs. pressure.

Yes, this is all described in Section 7 of the revised supplementary information.

However, it isn't easy to understand how the core composition modelling was done. One sentence was offered without further elaboration:

"Oxygen concentration depends on the concentration of sulfur in the core, the amount of FeO in the mantle, and the conditions of core formation Gendre et al. [2022]..."

This sentence is packed with unjustified assumptions. Another sentence:

"We, therefore, carried out multi-stage core formation models Badro et al. [2015] scanning all the possible values of input parameters (S content, FeO concentration, magma ocean geotherm, CMB pressure) and determined a range of oxygen concentrations as a function of sulfur concentration."

This procedure also needs to be better explained. The authors should give a lot more details on how the compositional models were developed.

This is a fair point. This is a whole other issue, and we realize this was awkwardly addressed in our paper. Accordingly, we rewrote a full section (Section S7) in the revised supplementary information, and described the whole core formation modelling in great detail.

We also re-organized and re-wrote more clearly the other sections related to core to remove redundancies. Now everything related to description of core formation modelling is packaged in the new Section 6.

The uncertainty in the alloys' EoSs is passed on to the core composition, which is modelled to fit the new Martian core model.

This philosophy is indeed correct, except it is done the other way around. A range of core compositions are first produced by running core formation simulations to find core compositions that match Martian mantle geochemistry (Supplementary Information Section 6). Then, the EoSs are run onto those models to keep only the ones that satisfy the geophysical inversion (Supplementary Information Section 7). The subset of core compositions obtained is therefore consistent both with core formation modelling (which is constrained by Martian mantle geochemistry) and core seismic properties (which are constrained by Martian core geophysics). We understand this confusion is our fault, because of the order in which we presented things in the original manuscript. Now we re-wrote these entire sections (see our answer to the previous comment). Section S6 deals with core formation modelling that comes first, and EoS constraints are in Section S7 and come after. We hope this clarifies our methodology.

The bottom line is: the EoS of iron and light element alloys produced in this paper supports the desired conclusion, i.e., the consistency with cosmochemical constraints.

Almost! As discussed above, it is the other way around, but it is the same equivalent. One can first constraint geochemistry, then geophysics, or the other way around. The result is, of course, the same.

I will not address other aspects of this paper since they fall outside my expertise.

Referee #3:

This paper explores an enriched partially molten silicate later at the base of Mars' mantle. The paper

presents a nice parallel approach for the two contrasting hypotheses: an inversion with and without the BML is pursued. In general, I find the paper interesting, but the seismic results seem a bit uncertain. Here are some comments/questions:

1. How do the ScS times behave (e.g., if differenced from SS times) for the two models?

Figure 3 displays the ScS-P travel times for both non-BML (panel a) and BML (panel b) models and observations. The agreement on $t_{ScS}-t_P$ between observations and the models for both BML and non-BML models is comparable, because, as discussed in the manuscript, from the point of view of seismic travel times one cannot distinguish between S-waves reflected at the top of a molten silicate layer (for BML models) and S-waves reflected at the top of a liquid core of larger size (for non BML models).

2. In considering parsimonious solutions, I would've thought a first approach would be to keep the core the same radius, and partially melt some portion of the lowermost mantle. Could that be done to fulfill differential times, yet still allow ScS to get to the CMB? If the answer is yes, then how is that hypothesis more or less compatible with other data? Wouldn't such a structure result in a negative velocity gradient down to the CMB (but not as severe as models in Figure 2), so wave paths will be strongly affected (and thus times too)?

While we appreciate the point made by the reviewer, having a metallic core of same size as earlier estimates overlain only by a partially molten silicate layer would be problematic for the following reasons:

(i) Having a partially molten mantle just above the CMB in a compositionally homogeneous mantle is difficult because the gradient of the solidus curve is steeper than the adiabatic mantle (see Fig. 1c). Hence, at the present-day one would need a strong temperature jump, ΔT_{CMB} of several hundreds of Kelvins in the lowermost mantle to reach super-solidus temperatures (Fig. 1c). Such a large temperature increase above the CMB, ΔT_{CMB} , cannot be obtained in a homogeneous Martian mantle at the present day, even in the bottom thermal boundary layer because ΔT_{CMB} decreases rapidly with time in a Martian mantle. This has been shown in various earlier studies (see *e.g.*, [Breuer and Spohn, 2003; Williams and Nimmo, 2016]) and is invoked to explain the extinction of the thermally driven geodynamo in Mars (assuming a compositionally homogeneous mantle).

(ii) Mantle melting above the CMB could be reached if this region is enriched in HPEs, which leads to BML models. However, in this case our current models and previous work [Samuel et al., 2021] show that it would lead to the presence of both a partially molten part and a fully molten mantle above the CMB because of the development of thermal gradient within and above the BML (see Fig. 1i).

(iii) As mentioned in the abstract and introduction, keeping a large core size would imply a low core density, with high amounts of light elements at odds with experimental petrology.

If the reviewer meant to keep a smaller core size directly overlain by a mushy mantle, this (in addition to the difficulty in such occurrence for both compositionally homogeneous or heterogeneous mantles as discussed in points (i) and (ii) above) would reduce the amount of light elements in the core mentioned above, but would not be consistent with measured tides and the moment of inertia of the core [Le Maistre et al., Accepted in Nature, 2023] that require a large apparent core size (see new section S9).

Overall, the presence of fully molten and partially molten enriched mantle layers above a smaller metallic core appears to be the simplest way to simultaneously satisfy geophysical and geochemical data.

3. Am I correct to read a +/- 5 sec in reported PP and Pdiff times? (though, the time uncertainty of PP looks huge from Fig S1). If yes, that could be a 10 sec error, correct? That is a significant part of the signal in column 1 of Fig 3 c,e.

The uncertainty in Fig. S1 was not plotted properly. We modified this figure and added a complementary figure S2 based on a frequency-dependent polarization analysis that consolidates our first picks. In panels (c) and (e) of Fig. 3 the 10 sec error, as correctly understood by the reviewer, on differential travel times is shown by the pink areas. Despite this relatively large uncertainty, non-BML models are unable to match the $t_{PP} - t_{Pdiff}$ travel times.

4. There are no predictions shown for the seismic waves for the two models, and no ScS waves are shown. That makes confidence in the seismic results difficult because of these uncertainties

We have added synthetic waveforms computed using non-BML and BML inversion sets to the revised manuscript (see Fig. S4 in the Supplementary Information). The newly added Section 4 in the Supplementary Information discusses these waveforms and we show in particular that BML models can reproduce signals that are consistent with the arrival times picked on InSight's seismic record for deep reflected S-wave and deep and diffracted P-waves.

In summary, this is a good assessment of the two models. I am not convinced they are the only possibilities, and I struggle with the seismic uncertainties. But it is provocative possibility.

Reviewer Reports on the First Revision:

Referee #1:

Review of Evidence and implications of an enriched molten silicate layer above Mars' core

Based on the rebuttal and scanning the revised manuscript and supplementary information, I am happy with the extensive improvements the authors have made to an already great study.

However, I have one lingering question concerning my original question:

E. page 6: "However, the differences in V^* imply that the mantle becomes considerably more viscous with depth for BML models because the pressure and temperature gradients in the convecting mantle are similar for the output models of both inversion sets. Indeed, neglecting the relatively small temperature effect compared to that of pressure, and assuming a pressure increase of $\Delta P \sim 15$ GPa throughout the convecting mantle, Eq. 1 implies a lowermost to uppermost solid mantle viscosity ratio, $R \sim \exp [\Delta P V^*/(R T)]$, leading to $R \sim 50$ for non BML models, and $R \sim 5000$ for BML models."

This is all a bit confusing.

a. E^* being smaller is understandable on account on the lower present-day T for SML models. But why is V^* so much larger in the BML models?

The authors' answer included a re-iteration of how this V^* (activation volume) is the result of their data-driven inference process. However, my lingering question is whether the authors could provide a physical justification/intuition for the large difference in V^* between the BML and non-BML models.

Other than that, I am excited to see this multidisciplinary paper in print.

Referee #2:

The description of the core properties modeling has improved and it is clearer now what has been done. Because there is a great deal of extrapolation from measurement conditions to core conditions, the uncertainty on the extrapolated property might not be much smaller than the error derived from "adjusted" ab initio calculations. This approach does not seem necessarily superior to "adjusted" ab initio results. Nevertheless, there is no alternative left.

I trust the materials modeling component is reasonable.

Referee #3:

The revised manuscript is improved and addresses questions in my first review. Some minor comments:

I don't understand Fig 1(n): there should be reciprocity on either side of the "PcP" segment, but the P(diff) portions are different on either side of the paths in the raypath figure. The red path looks hand drawn. Is it? If yes, why not show the actual computed path?

Fig S11 did not come through on the PDF I downloaded for review

I feel the paper now adequately supports the hypothesis put forward.

Author Rebuttals to Second Revision:

Referee #1:

Review of Evidence and implications of an enriched molten silicate layer above Mars' core

Based on the rebuttal and scanning the revised manuscript and supplementary information, I am happy with the extensive improvements the authors have made to an already great study.

However, I have one lingering question concerning my original question:

E. page 6: "However, the differences in V^ imply that the mantle becomes considerably more viscous with depth for BML models because the pressure and temperature gradients in the convecting mantle are similar for the output models of both inversion sets. Indeed, neglecting the relatively small temperature effect compared to that of pressure, and assuming a pressure increase of $\Delta P \sim 15$ GPa throughout the convecting mantle, Eq. 1 implies a lowermost to uppermost solid mantle viscosity ratio, $R \sim \exp[\Delta P V^*/(R T)]$, leading to $R \sim 50$ for non BML models, and $R \sim 5000$ for BML models."*

This is all a bit confusing.

a. E^ being smaller is understandable on account of the lower present-day T for SML models. But why is V^* so much larger in the BML models?*

The authors' answer included a re-iteration of how this V^ (activation volume) is the result of their data-driven inference process. However, my lingering question is whether the authors could provide a physical justification/intuition for the large difference in V^* between the BML and non-BML models.*

We are sorry for having missed the reviewer's point in the first round of review. The rheology of planetary mantles is not straightforward to predict or to explain because of the remaining tradeoffs between poorly constrained quantities that can affect the rheological behaviour of the mantle. For example, an intrinsically more sluggish mantle can be explained by a drier mantle (Karato & Wu, Science, 1993) but also by larger grain sizes, which are not explicitly modeled in our inversions. Therefore, explaining/justifying further the rheological differences between the two inversion outputs would be too speculative, because so far we cannot constrain the associated quantities that could explain these rheology (water content, grain size, major element content, etc.) or break the existing tradeoffs between these quantities on the rheology. We can only defer this quest to a future study. We added a text in the last version of the revised Methods Section to underline this point.

Other than that, I am excited to see this multidisciplinary paper in print.

So are we! Thank you.

Referee #2:

The description of the core properties modeling has improved and it is clearer now what has been done. Because there is a great deal of extrapolation from measurement conditions to core conditions, the uncertainty on the extrapolated property might not be much smaller than the error derived from "adjusted" ab initio calculations. This approach does not seem necessarily superior to "adjusted" ab initio results. Nevertheless, there is no alternative left.

We disagree with these statements. Our EoS for the core is based on an extensive set of experimental data acquired at conditions below, within, and above those prevailing inside the core of Mars (as mentioned in the supplement of our manuscript). Therefore, **our EoS applied to Mars' core does not require extrapolation.**

In particular, the non-ideal solution model for Fe-S (S likely being the most abundant light element in the core) is based on 89 different density measurements (conducted in the range 2-43 GPa) and 28 acoustic velocity measurements (conducted in the range: 0-52 GPa) for 23 different sulfur concentrations (5-37.5 wt.%). Densities and acoustic velocities predicted by our thermodynamic model at the experimental conditions are well within the uncertainties of the latter. Using Monte Carlo error propagation, the uncertainty is respectively less than 1% and 2% for density and acoustic velocity.

Our thermodynamic model accurately summarizes the relevant available experimental data, and can predict with a high confidence level the thermodynamic properties of the whole core at conditions (composition, pressure, temperature) falling within the range covered by the experiments. For this reason, **our core model based on experiments is currently the only viable option to infer core composition.** As argued in our previous rebuttal letter, ad hoc models build from a small number of "adjusted" (e.g., Huang et al.; 2023) ab initio calculations, are unreliable because they individually deviate significantly from experimental data.

Finally, the EoS used in our core model for the remaining liquid state end-members do not require pressure extrapolation either (Fe up to 350 GPa; FeO up to 100 GPa; Fe-C up to 50 GPa; Fe-H up to 152 GPa).

I trust the materials modeling component is reasonable.

We are happy that our changes made to the SI clarified our approach.

Referee #3:

The revised manuscript is improved and addresses questions in my first review. Some minor comments:

I don't understand Fig 1(n): there should be reciprocity on either side of the "PcP" segment, but the P(diff) portions are different on either side of the paths in the raypath figure. The red path looks hand drawn. Is it? If yes, why not show the actual computed path?

We thank the reviewer for highlighting the potentially confusing way that the raypaths were presented in Fig. 1n. In this figure we chose to plot a single raypath despite the fact that the diffracting nature of the Pdiff means that energy is continuously radiated into the molten layer beneath and reflected back toward the base of the fully solid mantle, where it resumes its diffracting path. All these paths have identical travel times and the energy that travels along them constructively interferes to produce the prominent signal we analyze in the manuscript. Naturally, for a surface event, the inherent symmetry of the geometry implies that the ensemble of all these paths must be symmetric. However, only the single path whose "PcP" bounce is at the midpoint of the raypath will itself also be symmetric; on either side, it will be flanked by a distribution of other paths that will not be individually symmetric. The red path in Fig. 1 is one such non-symmetric path and is not hand-drawn. We showed a subset of other paths in the "Pdiff-PcP" ensemble in Fig. S4 and now point the reader to this figure for a better sense of the range of paths that are associated with the observed arrival.

We therefore have left the path displayed in Fig. 1n but added text to the figure caption to explain the asymmetry.

Fig S11 did not come through on the PDF I downloaded for review

This is hopefully fixed upon resubmission of the last version of our revised manuscript.

I feel the paper now adequately supports the hypothesis put forward.

Excellent news. Thank you.